# Chronological adhesive cardiac patch for synchronous mechanophysiological monitoring and electrocoupling therapy

Chaojie Yu[1,2], Mingyue Shi[1,3], Shaoshuai He[1,4], Mengmeng Yao [1], Hong Sun [5] ✉, Zhiwei Yue[5], Yuwei Qiu[1], Baijun Liu[1], Lei Liang[1], Zhongming Zhao[1], Fanglian Yao [1,2] ✉, Hong Zhang [1,2] ✉ & Junjie Li [1,2] ✉

With advances in tissue engineering and bioelectronics, flexible electronic hydrogels that allow conformal tissue integration, online precision diagnosis, and simultaneous tissue regeneration are expected to be the next-generation platform for the treatment of myocardial infarction. Here, we report a functionalized polyaniline-based chronological adhesive hydrogel patch (CAHP) that achieves spatiotemporally selective and conformal embedded integration with a moist and dynamic epicardium surface. Significantly, CAHP has high adhesion toughness, rapid self-healing ability, and enhanced electrochemical performance, facilitating sensitive sensing of cardiac mechanophysiology-mediated microdeformations and simultaneous improvement of myocardial fibrosis-induced electrophysiology. As a result, the flexible CAHP platform monitors diastolic-systolic amplitude and rhythm in the infarcted myocardium online while effectively inhibiting ventricular remodeling, promoting vascular regeneration, and improving electrophysiological function through electrocoupling therapy. Therefore, this diagnostic and therapeutic integration provides a promising monitorable treatment protocol for cardiac disease.

Myocardial infarction (MI) is a malignant disease induced by ischemic arterial occlusion that necessitates early detection and treatment due to its acute onset, complex complications, and high lethality[1,2]. Bioelectronic devices that track cardiac function in real time and provide targeted feedback therapy are urgently required to control disease deterioration and minimize fatal complications[3–5]. Recently, epicardium patches achieved synchronized electrical recording and stimulation to regulate arrhythmias through sophisticated component design[6–8]. Nevertheless, in vivo electrophysiological detection is susceptible to electromagnetic interference, causing signal artifacts that misinterpret cardiac function[9]. Together with the lack of profound repair and regeneration of ischemic and fibrotic lesions in bioelectronic devices, there is an urgent need to develop flexible platforms that can promote myocardial repair while intuitively and precisely monitoring diastolic-systolic mechanophysiological function[10–12]. Fortunately, conducting polymer-based hydrogels have been demonstrated to not only responsively sense mechanical deformation but also compensate for electrical conduction in fibrotic tissues for electrocoupling treatment[13–15]. However, conventional conducting polymers depend on unsatisfactory hydrophobic processing, acid-doped environments, and invasive implantation[16–18]. Therefore, the molecular modification toolbox is expected to fabricate functionalized conducting polymers with hydrophilicity, electroactivity, and biocompatibility in a physiological environment[19],

[1]School of Chemical Engineering and Technology, Tianjin University, 300350 Tianjin, China. [2]Frontiers Science Center for Synthetic Biology and Key Laboratory of Systems Bioengineering, Ministry of Education, Tianjin University, 300350 Tianjin, China. [3]School of Chemical Science and Engineering, Tongji University, 200092 Shanghai, China. [4]Thrust of Sustainable Energy and Environment, The Hong Kong University of Science and Technology (Guangzhou), 511400 Guangzhou, China. [5]School of Basic Medical Sciences, North China University of Science and Technology, 063210 Tangshan, China. ✉e-mail: sunhong@ncst.edu.cn; yaofanglian@tju.edu.cn; zhanghong@tju.edu.cn; li41308@tju.edu.cn

bridging bioelectronics and tissue engineering to achieve reliable diagnostic and therapeutic integration.

Conductive hydrogels require strong tissue integration to ensure sensitive and reliable information communication and feedback for diagnostic and therapeutic purposes. However, conventional surgical sutures[20–22], microneedles[23–25], and bioadhesives[26–28] are nonconformable and unstable to the moist and dynamic myocardial surface, substantially diminishing the stress perception of myocardial mechanophysiology by epicardium patches and impeding electrocoupling in the bioelectronic–tissue interface[29–31]. The physical interactions[32–34] and chemical linkages[35–37] between the hydrogel and tissue can improve adhesion strength and interfacial toughness, but the limited functional groups in the epicardium surface result in low adhesive thresholds[38]. Additionally, the oil–water barrier on the epicardium surface suppresses the penetration and invasion of polymeric chain segments[39], which causes the construction of embedded topological connections and mechanical interlocking networks between hydrogels and myocardium to be a significant challenge. Critically, to prevent nonspecific adhesion of hydrogel patches to other tissues[40–42], there is an urgent requirement to modulate the gelation process, network structure, and viscoelasticity on the time scale to achieve spatiotemporally targeted and robust adhesion.

In this work, we engineer a chronological adhesive hydrogel patch (CAHP) that is conformably and directionally attachable to the moist and curvilinear epicardium for the diagnostic and therapeutic integration of MI disease (Fig. 1). Critically, functionalized polyaniline (f-PANi) is synthesized by a side-chain modification strategy with electroactivity and enhanced hydrophilicity in physiological media, considerably increasing the potential of conducting polymers for biomedical applications. The borate and carboxyl side chains in f-PANi spontaneously generate dynamic covalent borate ester bonds and noncovalent hydrogen bonds with polyvinyl alcohol (PVA), inducing the in situ formation of CAHP without additional stimulation in a physiological environment. When the precursor solution of CAHP is painted on the myocardial surface, amphiphilic f-PANi rapidly absorbs and removes the anti-adhesive pericardial fluid and voluntarily penetrates the epicardium. The synergistic chemical and physical cross-linkings in the CAHP concurrently enhance the internal cohesion and interfacial interlocking to firmly anchor with the myocardium. The

programmed adhesion mechanism based on proactive diffusion, interfacial cross-linking, and mechanical interlocking allows CAHP to progressively form a strong bridging network with the epicardium in contact during the initial gel state. Additionally, after complete gelation, the CAHP resists attachment to contralateral nontarget tissues. Compared with other two-part adhesive hydrogels[13,30], CAHP selectively forms embedded interlocking adhesion structures with tissues by simple manipulation of gelation time. Owing to the stable and compliant connection of the hydrogel–tissue interface and the dynamic and robust mechanical–electrical networks, the CAHP responsively translates diastole-systole-mediated microdeformations into sensitive resistive signal changes. Consequently, the CAHP-based flexible sensing platform enables in situ continuous recording of cardiac mechanophysiology to monitor the stroke amplitude and rhythm in abnormal hearts with MI. Significantly, CAHP can increase cardiomyocyte calcium transient (CT) velocity and improve MI-induced ventricular electrophysiological dysfunction by compensating for electrical conduction in fibrous tissue. Thus, CAHP synchronizes cardiac mechanophysiological diagnosis and electrical coupling therapy, offering promising MI treatment options.

## Results
### Structure and properties of functionalized polyaniline
f-PANi with a conjugated backbone and anionic side chain was synthesized by oxidative polymerization of three aromatic monomers: aniline, 3-aminobenzeneboronic acid, and 3-aminobenzoic acid (Fig. 2a). Compared with polyaniline (PANi), the stretching vibration absorption bands of B–O at 1340 cm$^{-1}$ appeared in both borated polyaniline (b-PANi) and f-PANi, suggesting the presence of borate side chains (Fig. 2b). The emerging asymmetric vibration absorption bands of C=O at 1710 cm$^{-1}$ in f-PANi proved the introduction of carboxyl side chains. Meanwhile, the O1s (531.8 eV), B1s (188.0 eV), and C=O (289.0 eV) peaks in the X-ray photoelectron spectrum (XPS) verified the side-chain structure of f-PANi (Fig. 2c, d). The percentages of N1s, B1s, and C=O peaks in f-PANi were 9.75%, 2.85%, and 4.95%, reflecting the presence of 29.23% borate units and 50.77% carboxyl units in the copolymer (Supplementary Table 1).

The zeta potential of PANi was +12.4 ± 0.5 mV, while the introduction of the anionic side chain reversed the zeta potential of b-PANi

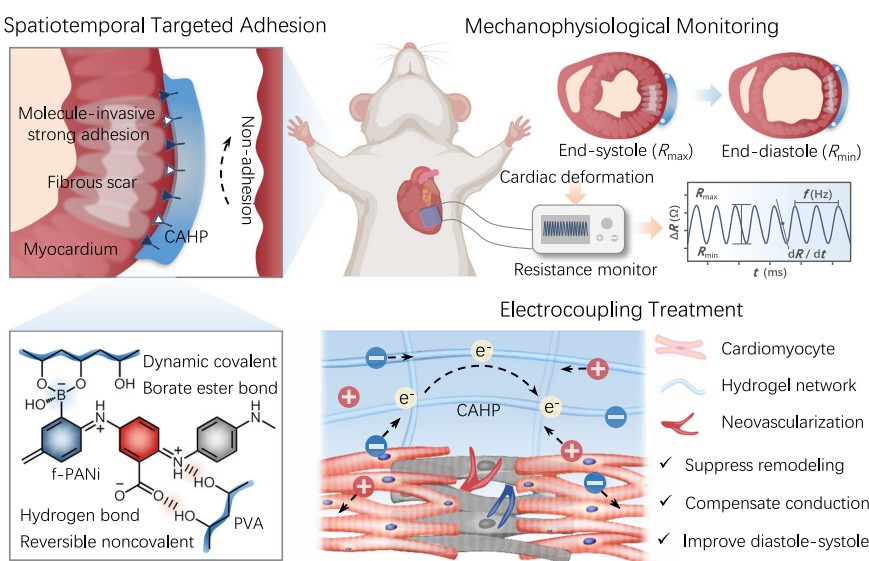

**Fig. 1 | Schematic illustration of a chronological adhesive hydrogel patch (CAHP) for synergistic cardiac mechanophysiological monitoring and electrocoupling therapy.** Chronological adhesion is mediated by interfacial dynamic covalent/noncovalent interactions between functionalized polyaniline (f-PANi) and polyvinyl alcohol (PVA), achieving molecule-invasive strong adhesion to the myocardium while resisting adhesion to nontarget tissues. CAHP can be used for cardiac mechanophysiological monitoring and simultaneous electrocoupling treatment of MI in rats.

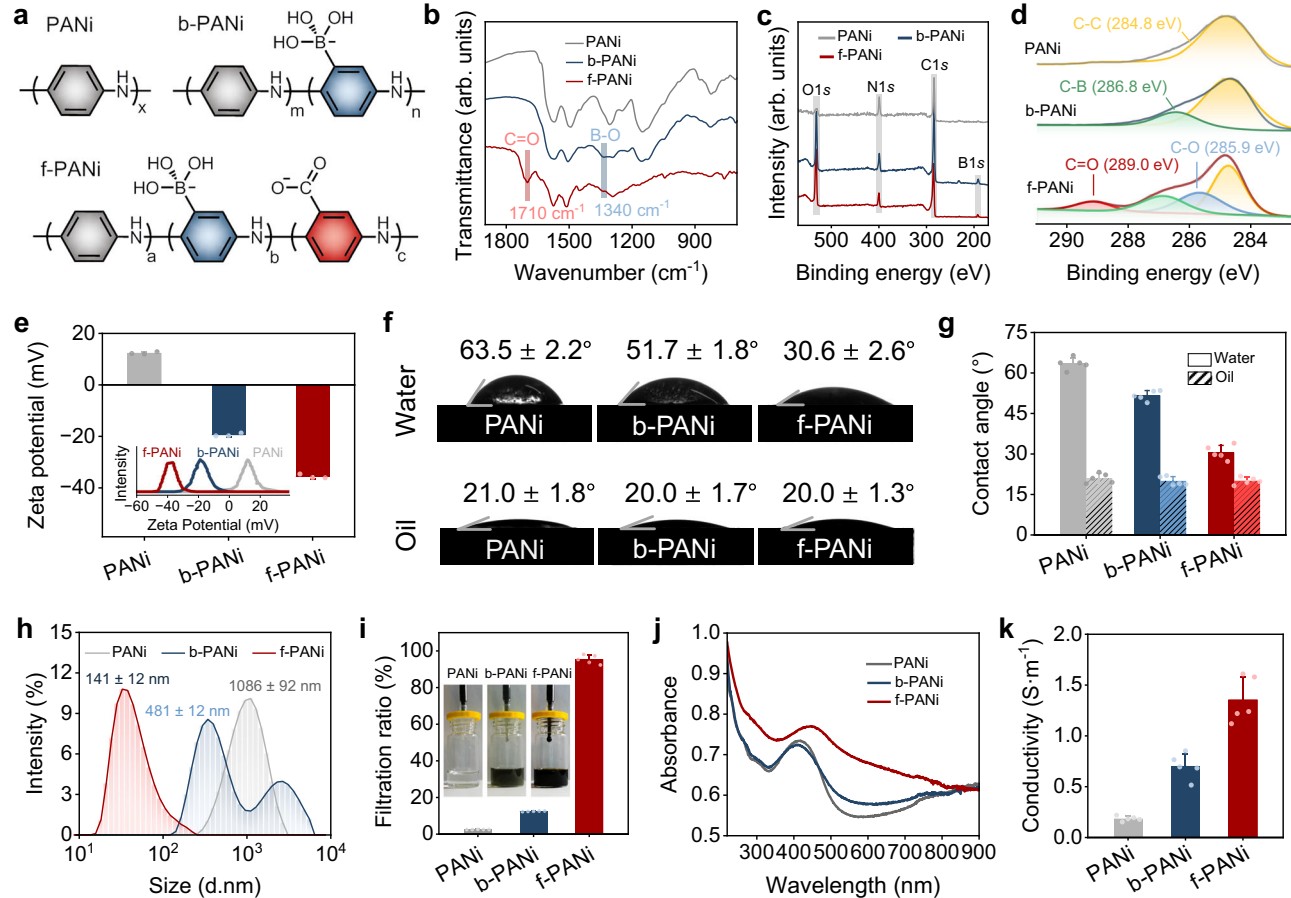

**Fig. 2 | Structure and properties of f-PANi. a** Chemical structures of the conducting polymers, including PANi, b-PANi, and f-PANi. **b** FTIR spectra of PANi, b-PANi, and f-PANi. **c** XPS survey for PANi, b-PANi, and f-PANi. **d** High-resolution spectra of C1s for different conducting polymers. **e** Zeta potential of conducting polymer aqueous dispersions ($n = 3$ independent experiments). **f** Diagram of water and oil droplets wetting conducting polymer films. **g** Water and oil contact angles of PANi, b-PANi, and f-PANi ($n = 5$ independent experiments). **h** The particle size of different conducting polymers dispersed in water. **i** Filtration ratio of PANi, b-PANi, and f-PANi. Inset: Photographs of conducting polymer aqueous dispersions passed through a membrane filter ($n = 5$ independent experiments). **j** UV–vis absorbance spectra of PANi, b-PANi, and f-PANi aqueous dispersions. **k** Conductivity of PANi, b-PANi, and f-PANi ($n = 5$ independent experiments). PANi: polyaniline, b-PANi: borated polyaniline, f-PANi: functionalized polyaniline. Data are presented as the mean ± standard deviation in (**e, g, i, k**).

($-19.4 \pm 0.6$ mV) and f-PANi ($-35.6 \pm 0.8$ mV) to negative values (Fig. 2e). Furthermore, the absolute value of the zeta potential of f-PANi was greater than that of PANi and b-PANi, suggesting that f-PANi showed stronger electrostatic repulsion and better stability in the aqueous system. Due to the conjugated backbone structures, conducting polymers had good lipophilic properties with an oil contact angle of approximately 20°. When the borate and carboxyl side chains were covalently grafted to the hydrophobic PANi backbone, the water contact angle decreased from $63.5 \pm 2.2°$ for PANi to $51.7 \pm 1.8°$ for b-PANi and $30.6 \pm 2.6°$ for f-PANi (Fig. 2f, g). The hydrophilic f-PANi could resist aggregation through electrostatic stabilization, overcoming the defect that conventional PANi was difficult to disperse in water. Consequently, the hydrated particle size of f-PANi was $141 \pm 12$ nm, which was significantly smaller than that of PANi ($1086 \pm 92$ nm) and b-PANi ($481 \pm 12$ nm; Fig. 2h). The f-PANi solution (12 wt%) could permeate through the filter membrane without resistance, whereas the PANi and b-PANi water dispersions (1 wt%) were difficult to filter out (Supplementary Movie 1). The filtration ratio of f-PANi ($95.4 \pm 2.4\%$) was dramatically higher than that of PANi ($2.6 \pm 0.1\%$) and b-PANi ($12.5 \pm 0.2\%$; Fig. 2i).

The conductive state of conventional PANi depends on the acidic environment and small molecule dopants, which are incompatible with the biosafety requirements in bioelectronics and tissue engineering. The borate and carboxyl side chains compensated for the charge on the conjugated structure. The UV absorption peak representing the polarization transitions was significantly red-shifted to 445 nm of f-PANi from 414 nm in PANi and b-PANi, which leads to easy electron excitation in f-PANi and enhanced mobility in the conjugated system (Fig. 2j). In addition, the relative intensity ratios of the quinone to the benzene ring in b-PANi and f-PANi were closer to 1, suggesting a redox intermediate state[43] (Supplementary Fig. 1). The FTIR absorption band of the benzene ring in f-PANi was blue-shifted to 1510 cm$^{-1}$ from 1493 cm$^{-1}$ in PANi, implying that f-PANi tends to have a more quinone-conjugated structure. Importantly, the content of protonated imines (−NH+=C−) in f-PANi (22.39%) and b-PANi (19.63%) was greater than that in PANi (11.85%), suggesting that the ionized side chains provide more protons for the imines in the backbone[44] (Supplementary Fig. 2a–d). Compared with PANi ($0.18 \pm 0.02$ S·m$^{-1}$) and b-PANi ($0.70 \pm 0.12$ S·m$^{-1}$), f-PANi has higher conductivity ($1.35 \pm 0.22$ S·m$^{-1}$) and is suitable for bioelectronic materials for electrophysiological diagnostic and therapeutic purposes (Fig. 2k).

## Chronological adhesive properties and mechanisms of CAHP
Mixing f-PANi and PVA solutions could induce the in situ formation of CAHPs in neutral media and at physiological temperatures. The borate groups on f-PANi could spontaneously form dynamic covalent borate ester bonds with the abundant hydroxyl groups on PVA. Furthermore, the carboxyl/imine groups on f-PANi could generate

hydrogen bonding interactions with PVA, synergistically modulating the mechanical properties. To demonstrate the relationship between the structure and properties of CAHPs, three CAHP samples with solid contents of 8 wt%, 10 wt%, and 12 wt% were prepared by adjusting the mass fractions of f-PANi and PVA solutions (Supplementary Table 2).

The polymer diffusion, network structure, and viscoelasticity of CAHPs were spatiotemporally specific during gelation, which supported the design of chronological adhesive hydrogels. The f-PANi and PVA solutions were mixed to gradually convert from the solution state (storage modulus $G' <$ loss modulus $G''$) to the hydrogel state ($G' > G''$), during which the CAHP was defined as the initial gel until $G' = G''$ and complete gel after modulus stabilization ($\Delta G < 1\%$). The increase in solid content increased the cross-linking density within the CAHPs, decreased the gelation times, and improved the mechanical modulus (Fig. 3a and Supplementary Fig. 3). Designing the reactive kinetics of borate esterification cross-linking on a time scale allowed

spatiotemporally specific tissue adhesion of CAHPs during their sol-gel transformation process (Fig. 3b). The CAHP was fluidized and applied to the tissue surface during the initial gel phase. Then, f-PANi rapidly crossed through the adhesion-resistant liquid membrane on tissue surfaces and penetrated the wet epicardium due to its modified hydrophilic properties. In the transition phase, diffusing f-PANi continuously formed click-type borate bonds with PVA at the hydrogel–tissue interface, which was the main covalent force for CAHPs to be firmly anchored to the wet tissue. The CAHPs achieved complete gelation after 10 min to produce dense cross-linking networks, whose pore sizes were approximately 10 times smaller than the initial gel state (Fig. 3c). The formation of tiny and dense pore channels at the tissue interface increased the specific surface area in contact with the tissue and enhanced mechanical interlocking adhesion. In the complete gel state, the ability of f-PANi to immerse into the gelatin substrate was significantly lower than that in the initial gel state (Fig. 3d and Supplementary Fig. 4). Therefore, the compact cross-linking in the

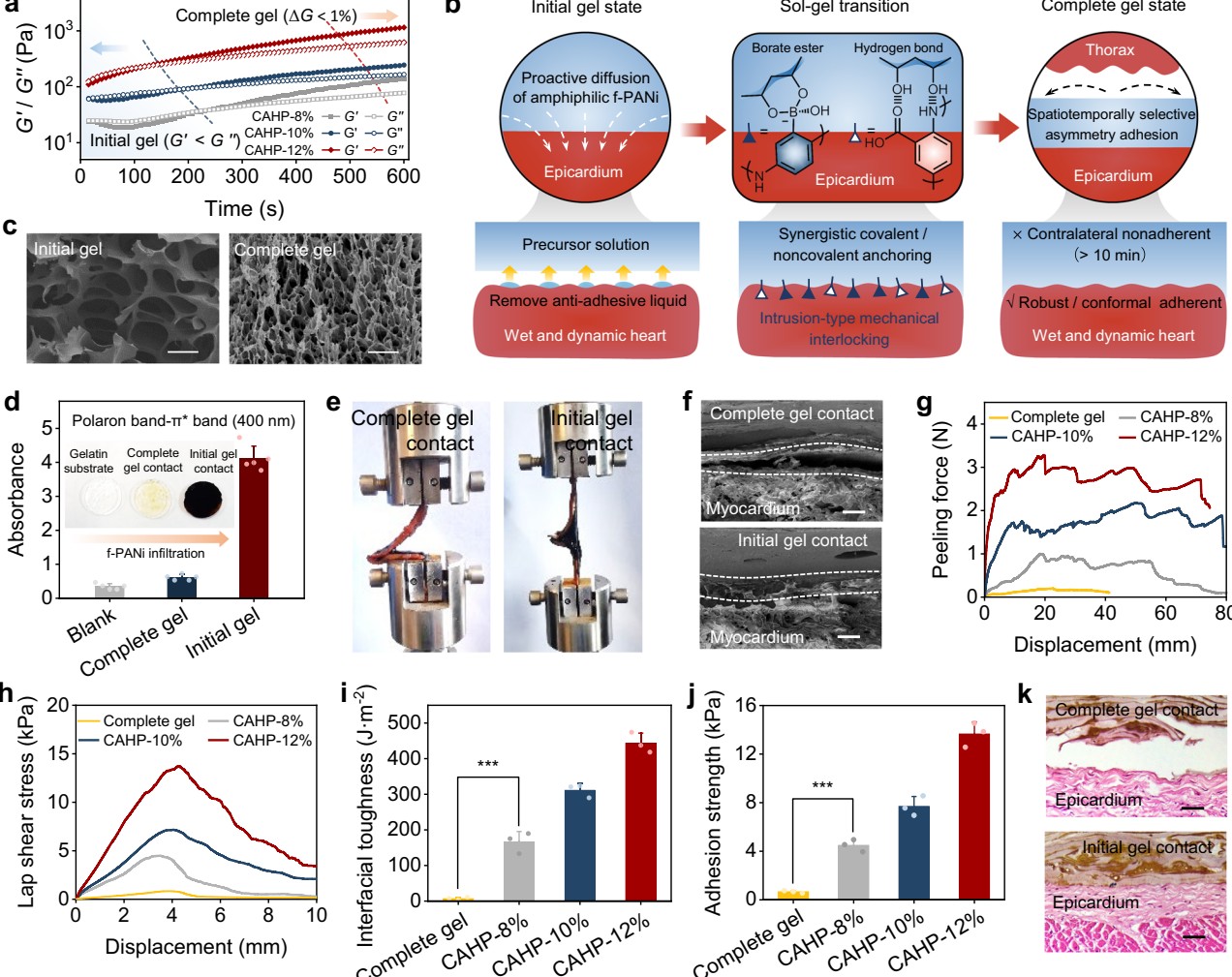

**Fig. 3 | Chronological adhesion performance and mechanism of CAHP.**
**a** Storage modulus ($G'$) and loss modulus ($G''$) variation of CAHPs during hydrogel gelation. **b** Chronological adhesion mechanism of the CAHP. **c** SEM images of the network structure of CAHPs in the initial and complete gel states. Scale bars, 100 μm (left) and 5 μm (right). **d** Absorbance at 400 nm of f-PANi infiltrated gelatin substrates during the initial and complete gel states ($n = 5$ independent experiments). **e** Peeling adhesion photographs when CAHPs in the initial and complete gel states are in contact with porcine myocardium for 10 min. **f** SEM images of the hydrogel–tissue interface. The white dotted line outlines the cross-section between the epicardium and CAHP. Scale bars, 20 μm. **g, h** Representative

force–displacement curves for hydrogel–tissue hybrids in peeling (**g**) and lap-shear tests (**h**). **i, j** Interfacial toughness (**i**) and adhesion strength (**j**) between CAHPs and porcine myocardium ($n = 3$ independent experiments).
**k** Hematoxylin-eosin-stained images at the contact interface between CAHPs and the epicardium. Scale bars, 20 μm. CAHP: chronological adhesive hydrogel patch. The measurements in (**c, f, k**) were repeated three times independently with similar results. Data are presented as the mean ± standard deviation in (**d, i, j**) and were analyzed using one-way ANOVA with Tukey's post hoc test in (**i, j**),
***$p < 0.001$. **i** $p = 7.34 \times 10^{-4}$ (Complete gel vs CAHP-8%). **j** $p = 2.04 \times 10^{-4}$ (Complete gel vs CAHP-8%).

complete gel phase restricted polymeric diffusion to avoid undesirable adhesion to nonspecific tissues.

To quantitatively analyze the chronological adhesive properties, peeling and shear adhesion experiments were used to test the adhesion strength and interfacial toughness of CAHP to porcine myocardium. The CAHPs were loaded between two layers of wet myocardium in the initial gel and complete gel states and incubated for 10 min. The initial gel allowed CAHP to conformably and firmly adhere to the epicardium after curing through polymeric penetration and an embedded interlocking mechanism (Fig. 3e, f and Supplementary Fig. 5), effectively resisting detachment of the adhesive interface during 180° peeling operations. However, the CAHP in the complete gel state had a smooth and moist surface, making it difficult to adhere to tissue even after prolonged contact. The fully cross-linked polymer network was unable to break through the anti-adhesive liquid film, separating the CAHP from the myocardial interface. The complete gel contact produced minimal peeling and shear forces, similar to a slight frictional effect (Fig. 3g, h). The interfacial toughness (166.2–443.4 J·m$^{-2}$) and adhesion strength (4.84–13.65 kPa) of the initial gel contact were significantly higher than those of the complete gel contact (7.39 J·m$^{-2}$, 0.66 kPa, Fig. 3i, j). When the solid content of CAHP was increased from 8 wt% to 12 wt%,

the ability to resist crack propagation under tension (Supplementary Fig. 6a–d) was improved, as evidenced by the increase in fracture toughness and fracture work (Supplementary Fig. 6e, f). Therefore, the increase in cross-linking density enhanced the ability to maintain interfacial interlocking and resist cohesive failure, resulting in an increase in the adhesion strength and toughness of CAHP by 2.67 and 2.82 times, respectively. Hematoxylin-eosin staining images showed that initial gel loading enabled CAHP to form an embedded inter-facial anchorage with the epicardium, whereas CAHP in the complete gel state no longer adhered to other tissues due to the absence of mechanical interlocking structures (Fig. 3k). This spatiotemporally specific programmed adhesive CAHP could adhere conformably and firmly to wet-state elliptical myocardial tissue and prevent undesirable adhesion of CAHP to the thorax or other organs.

## Customized paintable procedure of CAHP

The f-PANi and PVA solutions were homogeneously and thoroughly mixed to fabricate viscous and flowable initial gels using medical dispensers and syringes (Fig. 4a). The initial gels could be injected and shaped into hydrogel patches with uniform thickness by a custom applicator (Supplementary Fig. 7). This personalized manufacturing approach facilitated conformal integration between the paintable

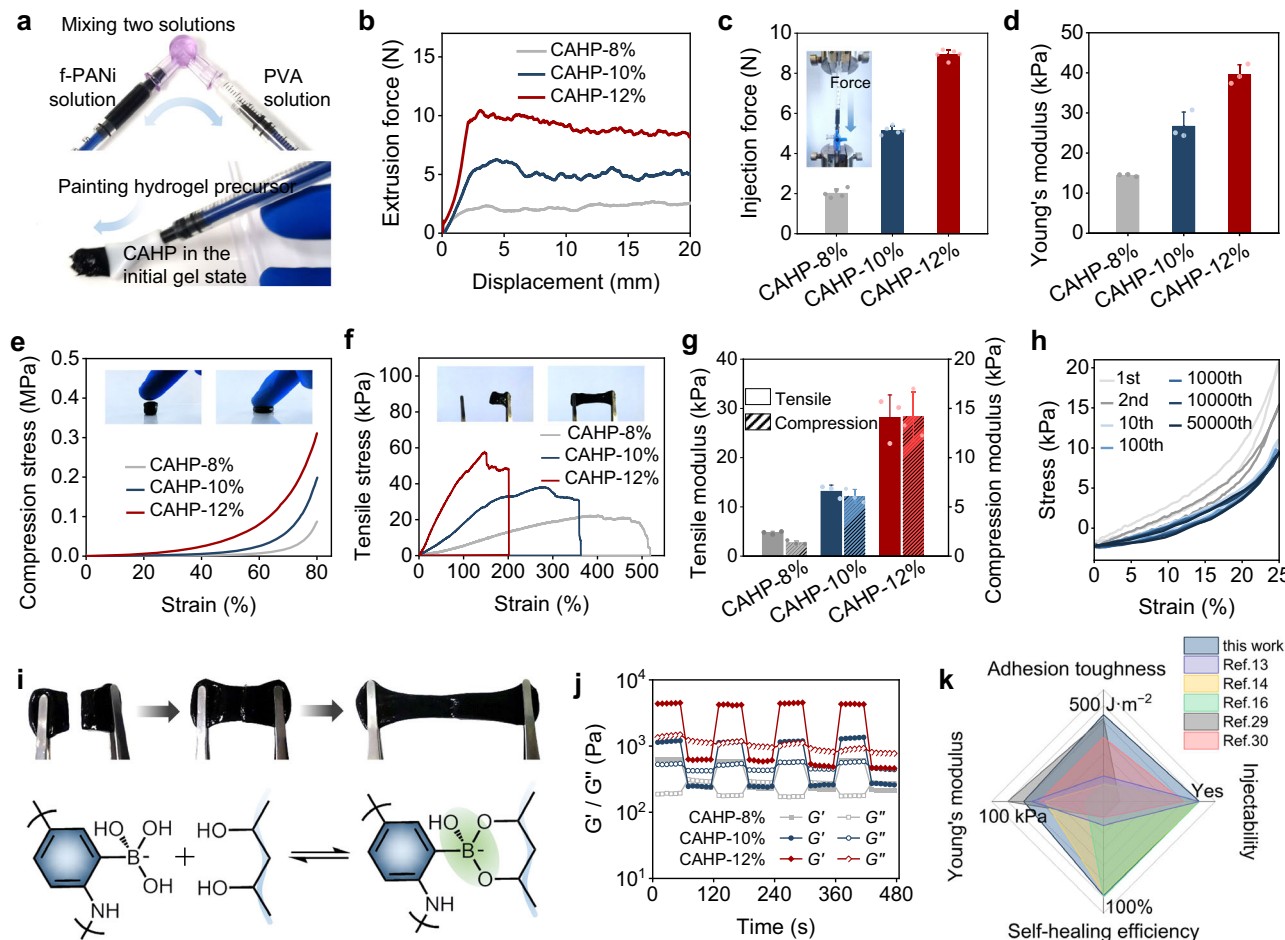

**Fig. 4 | Paintable ability, mechanical properties, and self-healing performance of the CAHPs. a** Photos of the preparation process and paintable performance of the CAHPs. **b** Representative extrusion force–displacement curve of the CAHPs. **c** Injection force of the CAHPs ($n = 5$ independent experiments). **d** Young's modulus of the CAHPs ($n = 3$ independent experiments). **e, f** Tensile (**e**) and compressive (**f**) stress–strain curves of the CAHPs. **g** Compressive and tensile modulus of the CAHPs ($n = 3$ independent experiments). **h** Representative loading–unloading tensile curves of CAHP-12% at a frequency of 1 Hz and deformation of 25% for

50,000 cycles. **i** The photos show that the CAHPs could rapidly self-heal and remain connected under stretching. Dynamic covalent borate ester bonds predominantly mediated the self-healing properties. **j** Modulus self-healing properties of the CAHPs when the alternate step strain was switched from 10 to 400%. **k** Comparison between the CAHP and other conducting polymer hydrogels[13,14,16,29,30] in terms of their injection, modulus, anti-fatigue, self-healing, and adhesion properties. PVA: polyvinyl alcohol, f-PANi: functionalized polyaniline, CAHP chronological adhesive hydrogel patch. Data are presented as the mean ± standard deviation in (**c, d, g**).

CAHP and ellipsoidal heart surface under minimally invasive surgery. The apparent viscosity of CAHP decreased with increasing loading shear rate, indicating shear-thinning properties to prevent blockage during injection (Supplementary Fig. 8a). Crucially, the CAHP recovered its viscosity level after unloading the shear, demonstrating that the reassociation capability of dynamic cross-linking avoided the depletion of mechanical properties by injection (Supplementary Fig. 8b). To assess the clinical manipulability of the paintable CAHP, a piston-compression experiment was performed to measure the force required for the CAHP to be injected at a rate of 5 mL·min⁻¹. On the force–displacement plot, the injection force increased rapidly during the initial phase and remained stable, representing the force that the physician initiated and sustained the plunger movement (Fig. 4b). All CAHPs generated less than 10 N injection force and were easily operated manually without additional equipment (Fig. 4c).

## Mechanical and self-healing properties of CAHP

The mechanical properties of CAHP could be regulated by adjusting the solid content. CAHPs isotonic with PBS showed insignificant swelling behavior and reached equilibrium within 12 h (Supplementary Fig. 9a). As the solid content was increased from 8 wt% to 12 wt%, the cross-linked density increased, and the equilibrium swelling rate decreased from $24.7 \pm 3.1\%$ to $11.4 \pm 2.5\%$ (Supplementary Fig. 9b). In the rheological frequency sweep measurements, the $G'$ of CAHP was consistently higher than the $G''$ in the angular frequency range from 1 to 100 rad·s⁻¹ (Supplementary Fig. 10a), suggesting that the CAHP maintained a stable hydrogel state in a dynamic environment that simulated cardiac deformation (20%) and frequency (1–10 Hz). The CAHPs exhibited a Young's modulus of 10–40 kPa (Fig. 4d) and had mechanical compatibility with soft physiological tissues ($E < 100$ kPa)[45]. In addition, tensile and compression tests were performed to evaluate the mechanical strength of the CAHPs. In the stress–strain curves for uniaxial tension, the breaking elongation decreased by approximately 2.5 times when the solid content of the CAHP increased from 8 wt% to 12 wt%, while the tensile modulus increased from $4.7 \pm 0.3$ to $28.1 \pm 4.6$ kPa (Fig. 4e, g). The compressive strength of CAHP at 80% strain improved with increasing solid content, and the compressive modulus was adjustable in the range of 1.4–14.2 kPa (Fig. 4f, g). In the cyclic tensile measurement at a frequency of 1.25 Hz and strain of 25%, CAHP-12% experienced stress decay and energy dissipation during the initial loading–unloading (Fig. 4h). The stress converged to a steady state after the tenth stretching, demonstrating that the CAHP could adaptively modulate the cross-linking network to accommodate the dynamic mechanical environment and exhibit resilience during simulated heart deformation.

The side-chain functionalization empowered the conducting polymer with synergistic reversible cross-linking capabilities, allowing the CAHP to be dynamically rearranged to recover the original percolation paths for electrical conduction after mechanical damage. Macroscopically, the separated CAHPs could be rapidly and autonomously connected upon contact (Fig. 4i), inhibit crack propagation, and enhance service life. Predominantly, dynamic covalent borate ester bonds can recombine without relying on external stimuli, mediating strong self-healing forces to quickly compensate for material defects. The severed CAHP self-healed quickly after re-contacting and maintained a stable connection after stretching (Supplementary Movie 2). Moreover, the reversible hydrogen bonding interaction between f-PANi and PVA contributed to cross-linked reconjugation. As the oscillatory strain increased from 10% to 1000%, the CAHPs underwent gradual cross-linkage decoupling and network collapse ($G' < G''$) from a mechanically steady ($G' > G''$) state (Supplementary Fig. 10b). In the step strain between 10 and 400%, CAHPs could be self-healed to their original modulus almost indestructibly (mechanical self-healing efficiency ≈ 98%) after stress damage, which was repeatable upon additional strain cycles (Fig. 4j). In addition, the electrical

self-healing efficiency in the disconnection-reconnection operation was approximately 100% (Supplementary Fig. 11). Compared to other self-healing conducting polymer hydrogels[46], f-PANi was able to directly participate in network remodeling, achieving higher mechanical and electrical self-healing efficiency. Comprehensively, compared with previous conducting polymer-based hydrogels incompatible with shear-thinning viscosity and supportable elasticity, the CAHP observably improved the mechanical properties of the self-healing network by integrating covalent and noncovalent bonds (Fig. 4k). In this regard, the CAHPs exhibited comprehensive advantages in injectability, self-healing, adhesion, and mechanical modulus, facilitating the manufacturing of minimally invasive, durable, and self-healing flexible electronics.

## Microstress sensing of CAHP

Maintaining tissue-adapted conductivity, stable charge injection, and storage in physiological media is fundamental to constructing electrodiagnostic and therapeutic bioelectronic devices. The CAHPs had a lower charge transfer resistance than the PVA hydrogels due to the electron conduction properties of f-PANi (Fig. 5a and Supplementary Fig. 12a), minimizing the interfacial impedance between the hydrogel and tissue. The electrical conductivity of CAHPs (1.05–1.38 S·m⁻¹) was an order of magnitude higher than that of PVA hydrogels (0.07 S·m⁻¹) and increased with the content of f-PANi (Fig. 5b). CAHP-12% exhibited stable conductivity during incubation in PBS at 37 °C for 30 days (Fig. 5c and Supplementary Fig. 12b), which was slightly higher than that of biological tissues (0.3–0.7 S·m⁻¹)[47], suggesting the ability to compensate for electrophysiological conduction in tissues. The charge storage and exchange capacity of CAHP was further characterized to assess sensitivity to exogenous electrical signals. In the cyclic voltammetry (CV) curve, the CAHP showed a characteristic oxidation potential (0.95 V) and reduction potential (−0.05 V) and a larger capacitance (1.451 mC·cm⁻²) than that of PVA hydrogels (0.527 mC·cm⁻²; Fig. 5d). Under electrical stimulation with biphasic pulses, the CAHP possessed a much higher charge injection capability (CIC) of 0.637 mC·cm⁻² than the PVA hydrogels (0.126 C·cm⁻²), indicating that the introduction of f-PANi improved the sensitivity of charge exchange (Supplementary Fig. 13a, b). Subsequently, the stability of charge injection was investigated, and the CIC value of CAHP-12% showed a decrease of less than 5% over 1000 charging and discharging cycles (Fig. 5e). Moreover, CAHP-12% showed negligible changes in CIC over 30 days of incubation in a physiological environment due to its good electrochemical stability (Supplementary Fig. 14a, b). Thus, the enhanced electrochemical properties have the potential to support low-noise, sensitive, and flexible sensing, simultaneously facilitating electrical coupling to repair damaged fibrotic tissues.

The dynamic mechanical network could quickly modulate the electronic pathway in response to loading micro deformation and stress. The CAHP associated with a resistance monitoring module could transform engineering strains into electrical signals for mechanophysiological detection. To investigate the application of CAHP in flexible mechanical sensing, uniaxial compressive (Supplementary Fig. 15a, b) and tensile tests (Supplementary Fig. 16a, b) were performed to examine the sensitivity of stress-induced current changes. As a result, the CAHP possessed high sensitivity ($k_{compression} = 2.10$ kPa⁻¹ and $k_{tensile} = 4.88$ kPa⁻¹) in the stress range of 0–15 kPa and decreased by a factor of 3.5 at stresses greater than 15 kPa (Fig. 5f, g). This result indicated that the CAHP was more sensitive to perceiving and responding to low stresses and was suitable for capturing some slight mechanophysiological motions in dynamic tissues. When the balloon model simulating the heart was ventilated and pressurized at a uniform rate, the pressure inside the balloon and the resistance of CAHP on the surface were simultaneously detected by a coupled pressure–resistance sensing system (Fig. 5h). The strong adhesion allowed the CAHP to mechanically integrate with the deformed surface

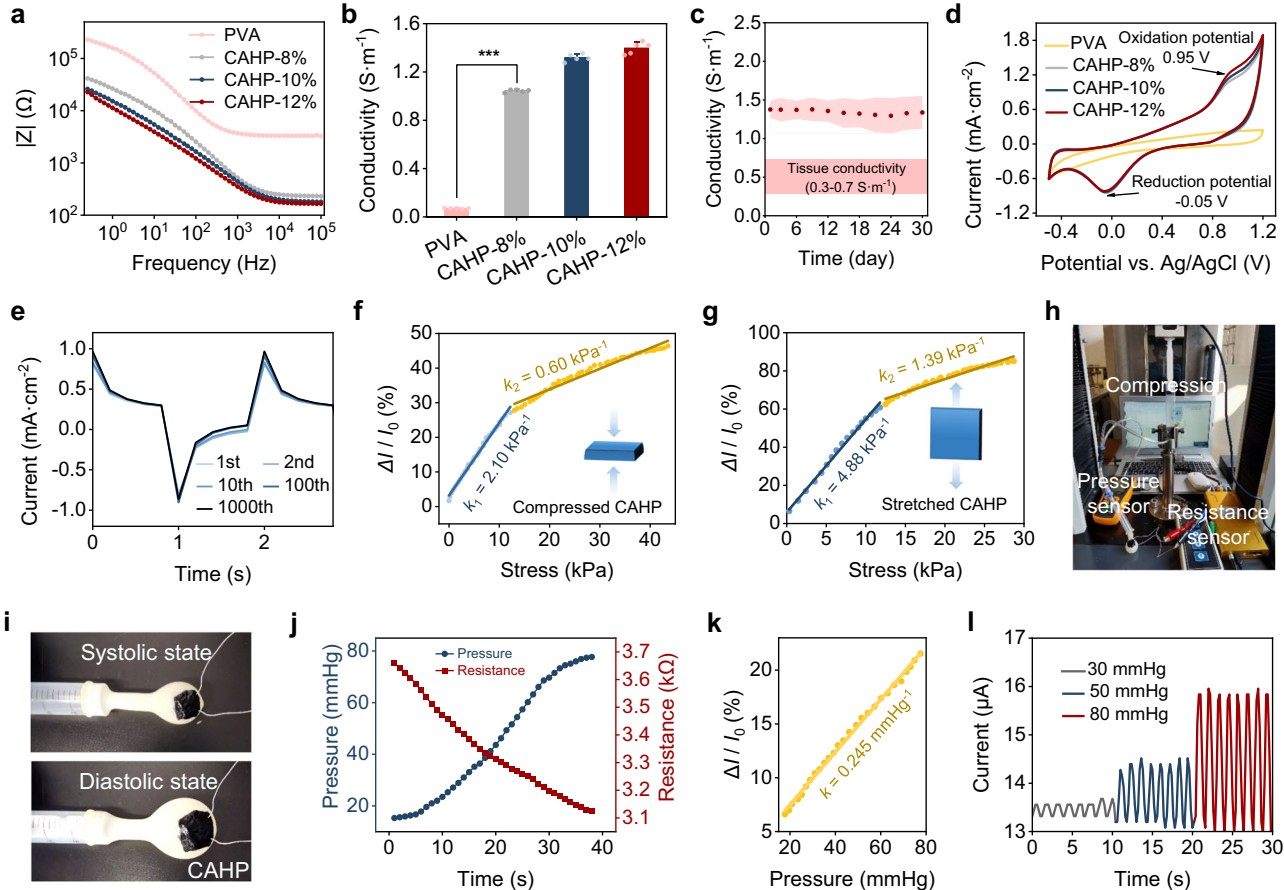

**Fig. 5 | Electrochemical properties and resistive sensing investigation of the CAHPs. a** Frequency-dependent impedance ($Z$) curves of the PVA hydrogel and CAHPs. **b** Conductivity of the PVA hydrogel and CAHPs ($n = 5$ independent experiments). **c** The conductivity of CAHP-12% incubated in PBS at 37 °C for 30 days ($n = 3$ independent experiments). **d** CV curves of the PVA hydrogel and CAHPs. **e** Charge injection curves of CAHP-12% with biphasic pulses of 1 s and ±0.5 V for 1000 cycles. **f, g** Uniaxial compressive (**f**) and tensile (**g**) stress-induced current changes ($\Delta I/I_0$) in the circuit. $k$ represents the sensitivity of stress-dependent current changes, which is the slope of the fitted curve. **h** Photograph of the pressure-resistance sensing system of the balloon model. **i** Photograph of the shape change of CAHP on the balloon surface under systolic and diastolic states. **j** Intraballoon pressure and interfacial CAHP resistance from model systole to diastole. **k** Sensitivity of CAHP-based resistive sensing in response to pressure changes in the balloon. **l** Current change in the CAHP when the internal pressure reaches 30 mmHg, 50 mmHg, and 80 mmHg. PVA: polyvinyl alcohol, CAHP chronological adhesive hydrogel patch. Data are presented as the mean ± standard deviation in (**b, c**) and were analyzed using one-way ANOVA with Tukey's post hoc test in (**b**), ***$p < 0.001$. **b** $p = 1.09 \times 10^{-15}$ (PVA vs CAHP-8%).

and responsively and reversibly generate volumetric and resistive variations. As the internal pressure increased from 20 to 80 mmHg, the balloon gradually changed from a systolic state to a diastolic state, inducing a linear reduction in the volume resistance of the CAHP and an increase in the internal current (Fig. 5i, j). Furthermore, the CAHP outputted current waveforms with different amplitudes at gradient pressures during multiple cycle strokes, and the sensitivity of the current sensing was 0.245 mmHg$^{-1}$ (Fig. 5k, l). The feasibility of CAHP for heartbeat-like monitoring is derived from three aspects: (1) conformal adhesion for precise compliance with guest deformation; (2) compatible dynamic and flexible mechanical properties for response to subtle stress changes; and (3) good electrochemical properties for high force-electric coupling capability and fidelity transmission of microcurrents.

## Cardiac mechanophysiological monitoring of the CAHP

Prior to the cardiac function monitoring study, the cytocompatibility and histocompatibility of CAHP were evaluated, demonstrating no effect on cell proliferation in vitro (Supplementary Fig. 17a, b) and little to no inflammatory response in vivo (Supplementary Fig. 18a, b). Subsequently, the CAHP with wires was conformally attached to the left ventricular surface in the live rat model by a chronological adhesion procedure (Fig. 6a and Supplementary Fig. 19). Then, the heart

deformation induced a cyclic change in the shape of the CAHP, which in turn varied the volume resistance and monitoring current in the CAHP (Fig. 6b). The resistance-type sensing module outputted a detectable current, which varied continuously over 3.9 to 15.0 μA. Concretely, the left ventricle at end-systole shrank in volume and pumped blood when the CAHP resistance was maximum, and the extracted current reached a lower limit (Fig. 6c). With progressive myocardial relaxation and ventricular filling, the CAHP resistance decreased, and the current increased until it reached a maximum at end-diastole. Therefore, the current value represented the deformation magnitude of the CAHP and indirectly reflected cardiac pulsation and pumping function, which has important applications in the diagnosis of MI. The MI model of the rat was constructed by ligating the left anterior descending branch, followed by installing a CAHP device to monitor cardiac function for 28 days (Fig. 6d). Ischemic myocardial injury induced myocardial cell apoptosis, electrophysiological dysregulation, and decreased heartbeat amplitude. As a result, the monitored current range (2–3 μA) decreased after 3 days of MI compared to the normal heart. The monitored current intensity ($\Delta I$) on day 14 and day 28 increased by 23.6% and 74.8%, respectively, compared with that at 3 days after MI, suggesting increased myocardial contraction amplitude and pumping volume (Fig. 6e). The maximum current change rate (max d$I$/d$t$) increased

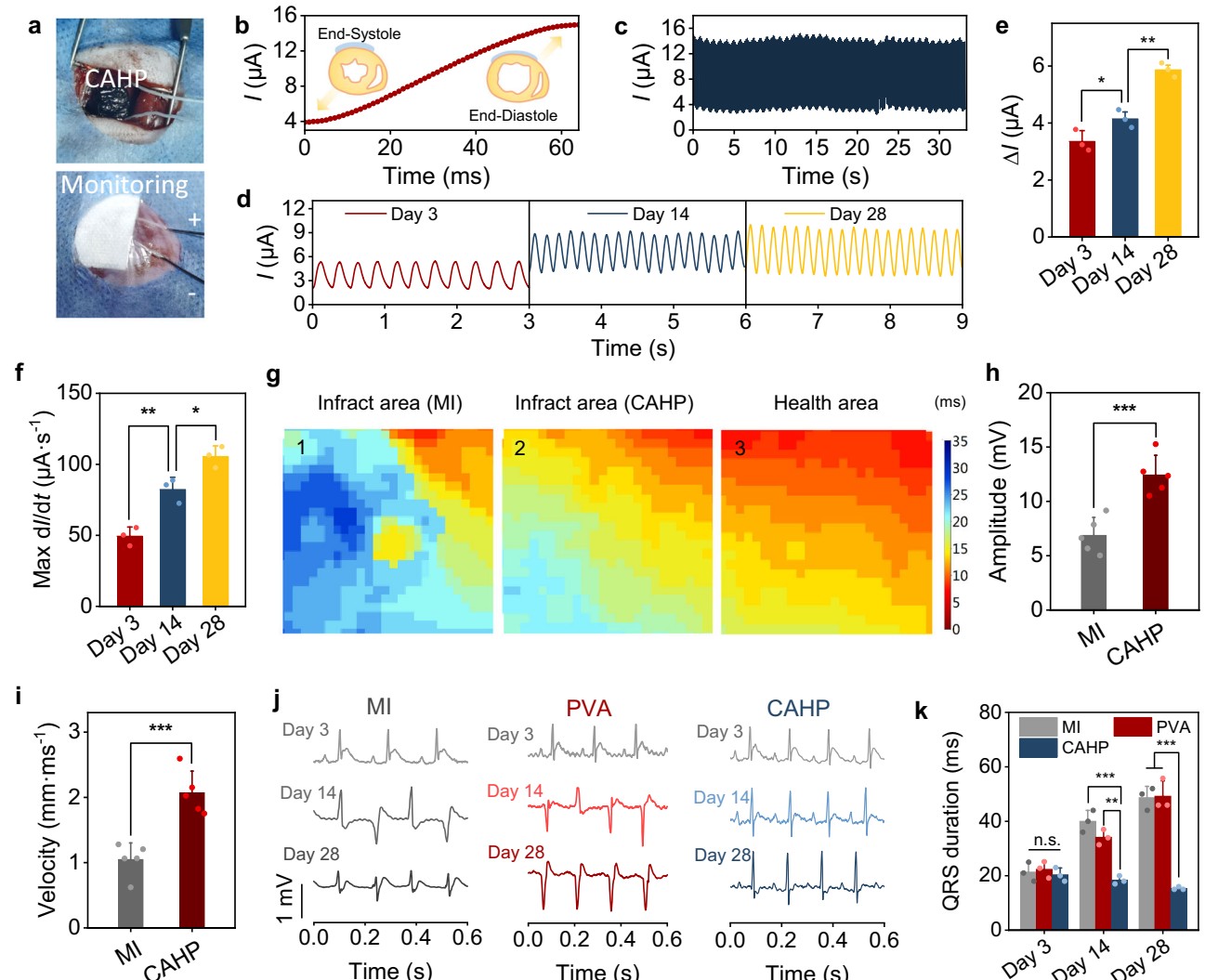

**Fig. 6 | Cardiac mechanophysiology monitoring and electrocoupling treatment by CAHP. a** Photographs of CAHP affixed to the left ventricular surface and mechanophysiological monitoring. **b** Current variation in CAHP from cardiac end-systole to end-diastole. **c** Current variation curve during CAHP monitoring of healthy hearts. **d** Current curves in the CAHP were recorded on days 3, 14, and 28 after MI. **e** The difference ($\Delta I$) between the peak and minimum currents ($n = 3$ animals). **f** The maximum value (Max d$I$/d$t$) of current variation in each stroke ($n = 3$ animals). **g** Heatmap of the activation time of calcium transient signals from the apices propagating to different regions of the heart. **h, i** Physiological potential amplitude (**h**) and average conduction velocity (**i**) in the MI and CAHP groups ($n = 5$ animals). **j** Representative ECGs for rats in the MI, PVA, and CAHP groups at 3, 14, and 28 days postoperatively. **k** QRS interval duration ($n = 3$ animals). MI myocardial infarction, PVA: polyvinyl alcohol, CAHP chronological adhesive hydrogel patch. Data are presented as the mean ± standard deviation and were analyzed using one-way ANOVA with Tukey's post hoc test in (**e, f, h, i, k**), n.s.: no significant difference at $p > 0.05$, *$p < 0.05$, **$p < 0.01$, ***$p < 0.001$. **e** $p = 0.0484$ (Day 3 vs Day 14), $p = 1.71 \times 10^{-3}$ (Day 14 vs Day 28). **f** $p = 6.04 \times 10^{-3}$ (Day 3 vs Day 14), $p = 0.0239$ (Day 14 vs Day 28). **h** $p = 9.90 \times 10^{-4}$ (MI vs CAHP). **i** $p = 6.12 \times 10^{-4}$ (MI vs CAHP). **k** Day 3: $p = 0.726$ (MI vs PVA), $p = 0.427$ (PVA vs CAHP), $p = 0.709$ (MI vs CAHP). Day 14: $p = 9.34 \times 10^{-4}$ (MI vs CAHP), $p = 1.05 \times 10^{-3}$ (PVA vs CAHP). Day 28: $p = 1.63 \times 10^{-4}$ (MI vs CAHP), $p = 5.03 \times 10^{-4}$ (PVA vs CAHP).

from $49.4 \pm 6.4$ μA·s$^{-1}$ to $105.5 \pm 7.5$ μA·s$^{-1}$, indicating that CAHP has the potential to improve the heart rate and pulsatile rhythm of infarcted myocardium (Fig. 6f). It is hypothesized that the electrical–mechanical coupling between the conductive hydrogel patch and the infarcted myocardium can be adapted to restore cardiac electrophysiology and pulsatile function[14,15]. Thus, the CAHP combines the dual functions of bioelectronic monitoring and tissue repair into an integrated diagnostic and therapeutic patch.

**Electrocoupling treatment of the CAHP**

To explain the electrical coupling treatment of CAHP, cardiac electrophysiology was visualized to analyze the compensatory electrical conduction effects of CAHP in the infarct region. The isolated perfused hearts were fluorescently labeled with CT dye and electrically stimulated at the apex. An optical mapping system recorded the electrical activation time and propagation process over the entire heart (Supplementary Fig. 20a). The left ventricular color in the MI group was blue, similar to the atrial color, representing a CT activation time of approximately 30 s. It was suggested that the severe fibrotic scar impeded electrical signal propagation. The left ventricle in the CAHP group was yellow–green, representing an activation time of approximately 15 s (Fig. 6g). The electrical coupling between CAHP and the infarcted myocardium promoted the electrophysiological activity of surviving cardiomyocytes in the infarcted area. The relative fluorescence intensity of the infarct region (location 1) in the MI group was significantly lower than that of the infarct region (location 2) in the CAHP groups and the healthy region (location 3), indicating that CAHP alleviated the electrical decoupling phenomenon of fibrotic tissue (Supplementary Fig. 20b). Additionally, CAHP decreased the calcium transient duration (CTD$_{90}$) from $98.5 \pm 3.8$ ms to $86.5 \pm 3.3$ ms

($p < 0.001$; Supplementary Fig. 20c) and increased the CT amplitude from $6.87 ± 1.67$ mV to $12.43 ± 1.81$ mV ($p < 0.001$; Fig. 6h), which was correlated with the elevated excitation-contraction coupling function. Importantly, the CAHP accelerated electrical propagation across the nonconductive scar tissue to the downstream and surrounding border area, obviously increasing the cardiac conduction velocity (Fig. 6i). Therefore, CAHP enhanced the electrical response and propagation and improved ventricular electrophysiological dysfunction induced by MI. To further verify the simultaneous treatment of MI by CAHP, an electrocardiogram (ECG) was recorded to validate the effect of CAHP on cardiac electrophysiology. On postoperative day 3, ECGs of all groups showed markedly elevated ST segments, suggesting acute myocardial injury (Fig. 6j). Subsequently, pathological Q-wave deepening, T-wave inversion, and prolongation of the QRS duration were observed in the MI and PVA groups, suggesting that electrically decoupled fibrosis affected depolarization and repolarization processes. CAHP attenuated ST-segment elevation, shortened QRS interval duration, and modulated cardiac electrophysiological behavior (Fig. 6k). This was because CAHP compensated for electrical conduction, activated the action potentials of cardiomyocytes, and alleviated MI-induced electrical decoupling.

### Myocardial repair effects of CAHP

Cardiac morphological staining further assessed the therapeutic effect of CAHP on myocardial repair. The MI group exhibited malignant ventricular remodeling, such as cardiomyocyte apoptosis, wall thickness thinning, and collagen deposition. The effect of the nonconductive PVA hydrogel on inhibiting infarct deterioration was limited. Compared to the MI and PVA hydrogel groups, the CAHP group had more myocardial fiber (red) survival and fewer collagen fibers (blue) (Fig. 7a). CAHP suppressed ventricular remodeling by mechanical compensation and electrical integration, significantly relieving ventricular thinning and maintaining wall thickness ($p < 0.001$; Fig. 7b). CAHP prevented further infarct expansion and fibrotic deposition, reducing the infarct area from $72.5 ± 8.6\%$ to $44.7 ± 8.1\%$ ($p < 0.001$; Fig. 7c). In addition, echocardiography was examined for cardiac geometry and function in the MI, PVA, and CAHP groups over 28 days postoperatively (Fig. 7a). MI results in significant ventricular dilatation, as evidenced by increased end-diastolic and end-systolic diameters and volumes, leading to progressive decreases in ejection fraction (EF), fractional shortening (FS), and stroke volume (SV). CAHP reduced the end-diastolic dimension and inhibited ventricular remodeling by mechanical support (Supplementary Fig. 21a, b). Furthermore, CAHP elevated electrically coupled contractility by compensating for electrical conduction in fibrous tissue (Supplementary Fig. 22a, b). In contrast, the PVA hydrogel was unable to assist myocardial systole-diastole because of the lack of conductive activity and was passive in limiting the geometric expansion of the infarcted myocardium. In addition, cardiac function tended to decline over 28 days in the MI and PVA groups, whereas CAHP improved cardiac pulsatile function. For example, the CAHP improved EF from $45.4 ± 2.9\%$ to $56.4 ± 2.4\%$ (Fig. 7d), FS from $23.2 ± 1.7\%$ to $31.0 ± 2.0\%$ (Supplementary Fig. 23), and SV from $107.4 ± 4.0$ μL to $119.0 ± 10.3$ μL (Fig. 7e), which were approaching the health value. Therefore, CAHP could restore cardiac function while being diagnosed in real time.

To explore the repair mechanisms, the infarct zones in the MI, PVA, and CAHP groups were analyzed by immunofluorescence staining. The CAHP group exhibited more extensive areas of positive cardiac troponin T (cTnT) and α-actin expression and more α-SMA-labeled vascular lumens (Fig. 7f and Supplementary Fig. 24a). In contrast, the PVA hydrogel had no significant modulating effect on vascular regeneration and electrical coupling. Notably, numerous vessels were observed in the epicardium surrounding the CAHP, restoring the blood supply in the ischemic region to prevent further disease

deterioration. Compared with the PVA group, the CAHP increased the blood vessel density from $6.1 ± 1.3\%$ to $10.2 ± 1.5\%$ ($p < 0.001$; Fig. 7g). The expression level of connexin 43 (Cx43) in the infarcted area was low in the MI and PVA groups, suggesting that intercellular electrical communication was greatly weakened. CAHP markedly improved Cx43 expression levels and promoted the myocardial electrical signaling pathway and electric contraction coupling (Fig. 7f and Supplementary Fig. 24b). In addition, cardiomyocytes have an inferior regenerative capacity and only compensate for quantity loss by enlarging their volume size. PVA hydrogel alleviated pathological myocardial hypertrophy through mechanical support ($p < 0.05$), whereas CAHP further reduced myocardial size from $502.1 ± 70.5$ μm$^2$ to $349.5 ± 68.0$ μm$^2$ ($p < 0.01$; Fig. 7f, h). Thus, compared with nonconductive PVA hydrogel, electrically coupled therapeutic CAHP could more effectively inhibit ventricular remodeling, reduce fibrotic scarring, promote vascular regeneration, and synergistically repair myocardial morphology and function.

## Discussion

We developed an electroactive and biocompatible f-PANi with ionized side chains and a conjugated backbone by a molecular modification strategy, which addresses the hydrophobic aggregation and decreased conductivity of conventional conducting polymers in physiological media. The borate-/carboxyl-mediated dynamic borate ester and noncovalent hydrogen bonds within f-PANi/PVA hydrogels promote self-healing mechanical networks and electrophysiological-adapted and charge-sensitive electrical networks. In addition, we propose a chronological adhesive scheme to achieve atraumatic, robust, conformable, and spatiotemporally selective integration of hydrogel patches into the moist dynamic epicardium.

This chronological adhesion is versatile and applicable to exploring widespread materials and medical applications. The conducting polymer-based CAHP enables simultaneous mechanophysiological monitoring and electrocoupling therapy, providing an all-in-one approach to diagnosing and treating MI. This medical-oriented CAHP can continue to serve other wearable and implantable electronic devices for health management, with the potential to repair other electroactive tissues, such as skeletal muscle and nerves. In the future, wireless, self-powered, and multimodal integration will further facilitate the clinical translation of this hydrogel-based bioelectronic diagnostic and therapeutic system.

## Methods

### Materials

Aniline (AN, 99.9%), 3-aminobenzeneboronic acid monohydrate (BAN, 98%), 3-aminobenzoic acid (CAN, 98%), and ammonium persulphate (APS, metals basis) were purchased from Aladdin. Polyvinyl alcohol (PVA, $M_w = 146–186$ kDa, 99% hydrolyzed) was purchased from Sigma-Aldrich. All reagents were used as received without purification.

### Synthesis of functionalized polyaniline

AN (2 mmol), BAN (4 mmol), CAN (8 mmol), and APS (14 mmol) were dissolved in 100 mL of HCl solution ($0.2$ mol·L$^{-1}$), and the polymerization was carried out at room temperature for 24 h. The f-PANi was purified by several aqueous washes and centrifugation (5000 rpm, 10 min) to remove impurities. The f-PANi solution was adjusted to neutral by adding an appropriate amount of NaOH solution ($3$ mol·L$^{-1}$), dialyzed for 3 days, and freeze-dried. The synthesis of PANi and b-PANi is described in the Supplementary Methods of the Supplementary Information file.

### Preparation and characterization of the CAHP

The f-PANi (8–12 wt%) was dissolved in PBS. The PVA solution (8–12 wt%) was obtained by dissolving PVA at 90 °C. Both solutions were sterilized by passing through a $0.45$ μm sterile filter. The CAHPs were prepared by

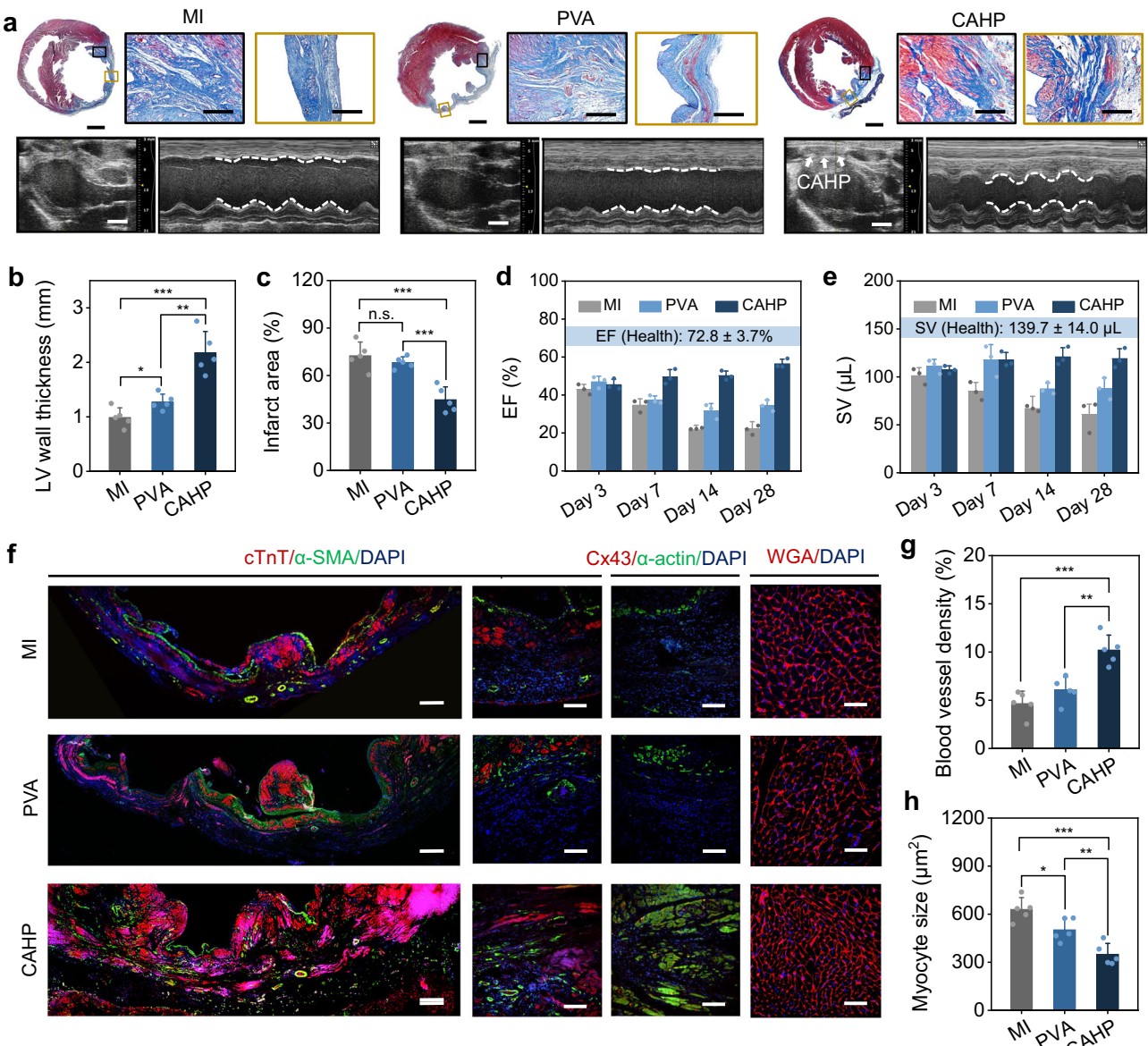

**Fig. 7 | Myocardial repair effects of CAHP. a** Masson staining for collagen (blue) and muscle (red) and echocardiography imaging in the MI, PVA and CAHP groups. White arrows represent CAHP on the epicardium, and the white dotted line outlines diastolic-systolic ventricular dimension changes. Measurements of Masson staining and echocardiography were repeated independently five and three times, respectively, with similar results. **b, c** Quantitative analysis of the wall thickness (**b**) and infarcted area (**c**) in the MI, PVA, and CAHP groups ($n = 5$ animals). **d, e** The ejection fraction (EF) (**d**) and stroke volume (SV) (**e**) of the left ventricle after 3, 7, 14, and 28 days postoperatively in the MI, PVA, and CAHP groups ($n = 3$ animals). **f** Representative immunofluorescence staining of the infarct region in the MI, PVA, and CAHP groups for cardiac markers (cTnT and α-actin), vascularization marker (α-SMA), intercellular electrical contraction coupling marker (Cx43), cardiomyocyte profiles (WGA), and nuclei (DAPI). **g, h** Quantitative analysis of blood vessel density (**g**) and myocyte size (**h**) ($n = 5$ animals). Scale bars, 2 mm in overall view, 300 μm in border and infarct area, and 4 mm in echocardiography (**a**); 300 μm (left) and 100 μm (right) in cTnT/α-SMA staining, 100 μm in Cx43/α-actin staining, and 50 μm in WGA staining (**f**). MI myocardial infarction, PVA: polyvinyl alcohol, CAHP chronological adhesive hydrogel patch. Data are presented as the mean ± standard deviation in (**b, c, d, e, g, h**) and were analyzed using one-way ANOVA with Tukey's post hoc test in (**b, c, g, h**), n.s.: no significant difference at $p > 0.05$, \*$p < 0.05$, \*\*$p < 0.01$, \*\*\*$p < 0.001$. **b** $p = 0.0241$ (MI vs PVA), $p = 1.29 \times 10^{-3}$ (PVA vs CAHP), $p = 2.66 \times 10^{-4}$ (MI vs CAHP). **c** $p = 0.332$ (MI vs PVA), $p = 3.51 \times 10^{-4}$ (PVA vs CAHP), $p = 7.67 \times 10^{-4}$ (MI vs CAHP). **g** $p = 2.74 \times 10^{-4}$ (MI vs CAHP), $p = 1.79 \times 10^{-3}$ (PVA vs CAHP). **h** $p = 0.0232$ (MI vs PVA), $p = 2.45 \times 10^{-4}$ (MI vs CAHP), $p = 8.27 \times 10^{-3}$ (PVA vs CAHP).

mixing the prepared f-PANi and PVA solutions in equal volumes using a connecting device. The detailed preparation parameters of the CAHPs are shown in Supplementary Table 2. Their morphologies were observed by scanning electronic microscopy (SEM, S-4800, Hitachi, Japan).

### Rheological measurements

Rheological experiments were carried out on a rheometer (Viscotester iQ, Thermo Fisher, USA) operating at 37 °C. Briefly, 400 μL of

CAHP precursor solution was loaded into a parallel plate. Time-sweeping measurements were conducted at a constant oscillation frequency of 1 Hz and an amplitude of 10%. The time corresponding to $G' = G''$ was defined as the gelation time. The yield strain of the CAHP was determined by strain sweep testing over a range of 10 to 1000% at a frequency of 1 Hz. The self-healing performance of the CAHPs was assessed by the time sweeping of alternating step strain. The angular frequency sweep was measured over a range of 0.5 to 100 rad·s⁻¹ with a constant amplitude of 5%. Young's modulus ($E$) was

calculated by Eq. (1):

$$E = 2\sqrt{G'^2 + G''^2} \cdot (1 + v) \tag{1}$$

where $G'$ and $G''$ are the storage modulus and loss modulus at 1 Hz, respectively, and $v$ is Poisson's ratio and is assumed to be 0.5.

The rotational testing of the rheology examined the paintable properties of the CAHPs. The hydrogel viscosities of the CAHPs were tested in the accelerated shear state (0.1–10 s⁻¹) and in the unloaded state (0.1 s⁻¹). In addition, time scans of the step shear rates were performed to assess the viscosity change during injection and painting with the shear rate switching between the low rate (0.5 s⁻¹) and high rate (5 s⁻¹) states with a duration of 60 s for each step.

## Adhesion measurements

The adhesion properties of CAHPs on chilled fresh porcine myocardium were detected using a universal machine (WDW-05, Si Pai Inc., China). In the lap-shear model, the tissue was cut into 10 × 26 mm rectangular sections using a razor blade, controlling the tissue thickness to 2 mm. Then, one side of the myocardium was glued to a glass slide (76 × 26 mm) with cyanoacrylate-based glue. The CAHP precursor solution was applied to a piece of tissue to trigger the hydrogel coating, onto which the other piece of tissue was pressed immediately or after complete gelation. The overlapping area was 10 × 26 mm, and the samples were allowed to cure for 10 min at 37 °C under humid conditions. Then, the samples were fixed between the two tensile fixtures and pulled apart at a velocity of 10 mm·min⁻¹. The adhesion strength was calculated by dividing the maximum shear force by the corresponding overlap area of each sample.

In the peeling test, the porcine myocardium was cut into 15 mm in width, 80 mm in length, and 2 mm thick pieces. The CAHP was adhered between the two myocardium at the initial gel or complete gel phase, with an overlapping area of 15 × 40 mm. The upper and lower fixtures clamped the two pieces of the myocardium. Then, the upper tissue was pulled upward at a constant velocity of 10 mm·min⁻¹ while the peeling force was recorded. The interfacial toughness was calculated by dividing twice the peeling force by the width of the tissue.

## Electrochemical measurement

The electrochemical properties of the CAHPs were examined at an electrochemical station (Vertex C, Ivium, The Netherlands). To measure electrical impedance, columnar CAHPs 15 mm in length and 9 mm in diameter were sandwiched between two copper electrodes with overlapping areas and equilibrated in Milli-Q water before testing. The two copper electrodes were connected to the electrochemical station to measure the electrical resistance. The conductivity ($\sigma$) of the CAHPs was calculated according to Eq. (2):

$$\sigma = \frac{d}{R \cdot S} \tag{2}$$

where $d$ and $S$ are the length and sectional area of the columned hydrogel samples, respectively. $R$ is the resistance at high frequencies.

The CV and CIC characteristics of the CAHPs were measured with a three-electrode system. A working electrode (the CAHP interface on a Pt sheet with an area of 1 cm × 1 cm), a counter electrode (Pt sheet, 1 × 1 cm), and a reference electrode (Ag/AgCl in saturated KCl solution) were immersed in a 10 mM PBS electrolyte. CV scans were conducted at a scan rate of 200 mV·s⁻¹ over a potential range of −0.5 to 1.2 V. The charge storage capacity ($Q_{stor}$) was calculated from the measured CV curves using Eq. (3):

$$Q_{stor} = \int_{E2}^{E1} \frac{I(E)}{2vA} dE \tag{3}$$

where $v$ is the scan rate, $E_2 - E_1$ is the potential window, $I$ is the current at each potential, and $A$ is the area of the hydrogel.

In the CIC measurements, electrochemical current pulse injection (biphasic pulses of 1 s and ±0.5 V) tests were performed for 1000 cycles in PBS solution using an electrochemical workstation. The charge injection density ($Q_{inj}$) was calculated from the measured charge injection curves using Eq. (4):

$$Q_{inj} = \frac{Q_c + Q_a}{A} \tag{4}$$

where $Q_c$ is the total injected charge in the cathode phase, $Q_a$ is the total injected charge in the anode phase, and $A$ is the area of the hydrogel.

To characterize electrical stability, the CAHPs were placed in PBS at 37 °C and 95% humidity for 30 days. The conductivity and charge injection density of the CAHPs were tested every 3 days.

## Microstress sensing tests

Tensile and compression tests were performed using a universal machine. Mechanical testing of CAHP was performed after swelling equilibrium in 10 mM PBS. For tensile tests, the CAHPs were made into rectangular shapes (gauge length of 10 mm, width of 10 mm, thickness of 2 mm) and stretched at a tensile rate of 50 mm·min⁻¹. Cylindrical CAHPs with a diameter of 9 mm and a height of 10 mm were used for the compression tests. The loading rate was fixed at 50 mm·min⁻¹, and the termination strain was 80%. The nominal stress ($\sigma$) was obtained by dividing the loading force by the cross-sectional area of the original specimen, and the nominal strain ($\varepsilon$) was defined as the change in length divided by the original length of the samples. The loading–unloading tensions were measured 50,000 times at 1 Hz over a 25% deformation range. The elastic modulus was calculated according to the initial linear slope of the stress–strain curve. The CAHPs were applied with a potential of 3 V during tension and compression, and the electrochemical workstation simultaneously recorded the stress-induced current change. The sensing sensitivity $k$ is calculated using Eq. (5):

$$k = \frac{d\left(\frac{\triangle I}{I_0}\right)}{dF} \tag{5}$$

where $\Delta I/I_0$ is the fraction of current change and $F$ is the stress.

The syringe-balloon model simulated ellipsoidal cardiac deformation. The CAHP (1 × 1 cm) was painted on the balloon surface, and the two ends of the CAHP were connected to the electrochemical workstation with silver wires. The balloon deformation was adjusted by controlling the syringe piston. The pressure transducer recorded the pressure changes inside the balloon, and the electrochemical workstation recorded the deformation-mediated current changes in the CAHP at an operating potential of 0.5 V.

## Myocardial infarction model construction

Animal experiments were approved by and performed according to the guidelines of the Institutional Animal Care and Use Committee of the North China University of Science and Technology. MI was induced in 8-week-old male Sprague Dawley rats. The rats were anesthetized with isoflurane, intubated, and ventilated (R145, RWD, China). Then, the heart was exposed via intercostal thoracotomy. The left anterior descending branch was ligated with a 6.0 suture at 2–3 mm between the pulmonary artery conus and the left atrium. Myocardial whitening in the downstream region confirmed successful ligation. MI-induced rats were randomly divided into the MI group ($n = 5$), PVA group ($n = 5$), and CAHP group ($n = 5$). Healthy SD rats ($n = 3$) were used as the control group. Three rats were randomly selected from each group and

underwent mechanophysiological monitoring, echocardiography and ECG at different times.

## Cytotoxicity and histocompatibility evaluation

The CAHP was immersed in a complete medium to obtain extracts, which cultured mouse fibroblast L929 cells (iCell-m026, iCell Bioscience) in 96-well tissue culture plates (Jet Biofil) for 24 h and 48 h, respectively. Cell viability was analyzed using the MTT cell proliferation kit (M1020, Solarbio) and enzyme immunoassay (Infinite M Nano+, Tecan, Austria). Cell morphology and number were observed using acridine orange staining (CA1143, Solarbio). The CAHPs were subcutaneously embedded in the subcutaneous tissue of Kunming mice (male, 8 weeks old, $n = 3$) for 7 days, and hematoxylin-eosin staining (DH0006, Leagene) of the hydrogel–tissue section was performed.

## CAHP implantation and mechanophysiological monitoring

The CAHPs were painted to the normal and infarcted left ventricular wall with a customized mold, and silver wires with insulation were mounted on both ends of the CAHP. After the complete gelation of the CAHP, the thorax was closed, and the wires were placed under the pectoralis. The skin was sutured and secured with a sterile patch. The wires were connected to an electrochemical workstation, a potential of 0.5 V was applied to the CAHP, and current change curves were recorded at 3, 14, and 28 days postoperatively.

## Electrocardiogram

On days 3, 14, and 28 after surgery, rats were anesthetized with isoflurane and fixed on the operating table. Needle electrodes were inserted subcutaneously into both the upper and right lower extremities. ECGs in the MI, PVA, and CAHP groups were acquired using a biosignal processing system (PowerLab System, AD Instruments, Australia). LabChart 7 software recorded the ECGs for 5 min at a tracking speed of 50 mm·s$^{-1}$ and determined the QRS interval duration.

## Echocardiography

Transthoracic echocardiography (Vevo 2100, VisualSonics, Canada) was performed to examine cardiac dimensions and functions. Two-dimensionally guided M-mode measurements captured parasternal short-axis views at the level of the midpapillary muscle. The ventricular end-diastolic diameter, end-systolic diameter, end-diastolic volume, end-systolic volume, FS, EF, and SV were measured on days 3, 7, 14, and 28 in the healthy, MI, PVA, and CAHP groups. All measurements were averages of three consecutive cardiac cycles.

## Electrophysiology mapping

On day 28 after surgery, the rats were euthanized under anesthesia. The isolated hearts in the MI ($n = 5$) and CAHP ($n = 5$) groups were perfused with excitation-contraction uncoupling reagent (ab120425, Abcam) via the Langendorff system to gradually stop beating but maintain electrophysiological activity. Perfusion of 50 μL of Pluronic™ F-127 (P3000MP, Thermo Fisher) into the heart increased cell membrane permeability. Intracellular calcium transients were fluorescently labeled by 100 μL of the calcium indicator Rhod-2 AM (ab142780, Abcam). An electrical stimulus at twice the pacing threshold current and 5 Hz pulses were executed at the apex and conducted toward the atria. The CT propagation signals were recorded by a fluorescent mapping system (OMSC801, MappingLab, England). The activation time, conduction velocity, amplitude, and CTD$_{90}$ were analyzed using OMapScope 5 software.

## Histopathological and immunohistochemical examination

On day 28 after surgery, the hearts in the MI, PVA, and CAHP groups were fixed with 4% paraformaldehyde, dehydrated, and embedded in paraffin. Heart specimens were sliced into 4 μm thick sections for histological staining. Myocardial morphology and fibrosis were assessed by Masson trichrome staining (DC0032, Leagene Biotechnology). Images were captured with a microscope (BX63, Olympus, Japan), and left ventricular wall thickness and infarct area were analyzed with ImageJ software. The infarct area was the area fraction of collagen fibers. α-Actin (M1206-1, Huabio, dilution 1:200), cTnT (ET1610-51, Huabio, dilution 1:200), and Cx43 (A11752, ABclonal, dilution 1:100) were selected as cardiac-specific markers for immunofluorescence staining to evaluate myocardial repair. α-SMA (904601, BioLegend, dilution 1:200) was selected as a vascular marker for immunofluorescence staining to detect revascularization. Wheat germ agglutinin (WGA, W11262, Thermo Fisher, dilution 1:400) and DAPI (8961, Cell Signaling Technology) were used to stain the cell membrane and nucleus. The fluorescent secondary antibodies were AF594 anti-rabbit (A21207, Thermo Fisher, dilution 1:500) and AF488 anti-mouse (A21202, Thermo Fisher, dilution 1:500). A high-speed confocal platform (Andor Dragonfly, Oxford Instruments, England) captured fluorescence images. ImageJ software was utilized to analyze blood vessel density, the relative fluorescence intensity of Cx43, and myocardial size.

## Statistical analysis

All experiments were performed at least three times ($n \geq 3$) for each sample. All data are presented as the mean ± standard deviation. Data were analyzed by Origin 2023 software and Microsoft Excel 2016 using one-way ANOVA with Tukey's post hoc test. A value of $p < 0.05$ was considered significant. n.s.: no significant difference at $p > 0.05$. Animal experiments were not considered for sex analysis because myocardial structure and functions in the MI model and inflammatory responses in subcutaneous encapsulation were not related to sex.

## Reporting summary

Further information on research design is available in the Nature Portfolio Reporting Summary linked to this article.

## Data availability

All data supporting the findings of this study are available within the paper and its supplementary information or from the corresponding authors on request. Source data are provided with this paper.

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

## Acknowledgements

The authors acknowledge P. Che and X. Jiang from the School of Basic Medical Sciences, North China University of Science and Technology, for help with pathomorphological tests and M. Sun from Basic Medical Research Centre, Tianjin Medical University, for help with echocardiography. The authors acknowledge support from the National Natural Science Foundation (U20A20261 and 52073205, J.L.) and Tianjin Research Innovation Project for Postgraduate Students (2022BKY091, C.Y.).

## Author contributions

C.Y., F.Y., and J.L. designed and conceptualized this study. C.Y., M.S., S.H., and M.Y. carried out the material characterization and animal experiments. H.S., Z.Y., and Z.Z. constructed the experimental animal model. Q.Y., B.L., and L.L. provided cell characterization protocols. F.Y., H.Z., and J.Li supervised the overall conception and design. C.Y., M.S., and H.Z. contributed visualization. C.Y. and J.L. wrote and revised the manuscript. All authors commented on the manuscript.

## Competing interests

The authors declare no competing interests.
