## [Peer Review File · Nature Communications]

Reviewers' Comments:

Reviewer #1:

Remarks to the Author:

The authors report a novel two-part tissue-adhesive conductive hydrogel-based cardiac patch for cardiac monitoring and electrocoupling treatment of myocardial infarction. The benchtop characterization and analysis of the material design, chemistry, adhesion performance, and electrical and electrochemical properties were conducted. The pre-clinical evaluation of the cardiac monitoring and electrocoupling treatment efficacy of MI was performed based on a small animal model. The proposed material and its application for in vivo cardiac monitoring and MI treatment were well characterized based on the comprehensive set of experiments and corresponding analyses for chemical, mechanical, electrical, and biological aspects of the material as well as the target application. The data and presentation are of high quality and comprehensive for the early R&D stage research. The reviewer finds the current work would be a meaningful addition to the community although there are several minor suggestions that the authors can address before the publication. The detailed comments are provided below:

1. The two-part injectable in situ cross-linkable tissue adhesives are widely studied and even commercialized in various compositions and target applications including epicardial interfacing. Although specific chemistry of materials might differ with the current work, the overall sol-gel transition (non-adhesive after gelation) and adhesion mechanism (initial physical contact of epicardial surface -> formation of covalent/physical bonds + mechanical interlocking upon crosslinking/gelation) of those existing solutions are essentially comparable to the current work. In this sense, the reviewer finds a bit confused about what the authors mean by 'chronological' and 'spatiotemporal' greatly emphasized throughout the work given it is really not really unique feature of the proposed solution. It would be great if the authors can clarify how the proposed solution may uniquely differ from existing two-part tissue adhesive hydrogels or make the discussion more technically focused than conceptual selling.
2. Based on Fig. 3E, the failure mode is cohesive failure. In this case, the adhesion performance may largely dependent upon the fracture toughness of the CAHP but it was not separately measured in the work. It would be informative to provide a fracture toughness of the CAHP as it may directly correlate with its adhesion performance.
3. It is unclear that whether all mechanical characterizations were conducted with as-prepared/applied CAHP or fully equilibrated (or in equilibrium swollen state in physiological environment) states. As the CAHP only contains dynamic covalent bonds (with relatively lower bond energy) and physical bonds, the stability and configurational changes (swelling, mechanical properties, etc) would be critical to actual in vivo application and efficacy. The authors may need to provide clarification on the test conditions as all of these tests would best be done in fully swollen/equilibrated conditions not as-prepared/applied states.
4. The authors correlate data in Fig. 4H to the anti-fatigue property of the CAHP. However, small hysteresis in cyclic tensile tests represents resilience, not anti-fatigue property as the authors seems didn't provide specific fatigue threshold data. Related, the data shows that the material does not have mechanical dissipation but it is confusing how the material can be tough & provide good adhesion performance. What's the exact mechanism of material robustness and good adhesion performance?
5. Regarding in vivo treatment of MI, the reviewer is curious whether the authors could decouple the mechanical support effect by adhering a swollen hydrogel to the epicardium. It has been known that mechanical augmentation by adhesive viscoelastic hydrogel to MI heart can provide a significant repair effect without electrical conductivity (for example, Lin et al, Nature Biomedical Engineering 3, 632-643 (2019)). A rodent MI model is susceptible to mechanical and electrical effect coupled due to small organ size potential volume-occupying effect of the sizable implant to the heart. The reviewer thinks that having a negative control group (similar hydrogel without conducting polymer) maybe helpful to highlight the electrocoupling treatment efficacy alone.

Reviewer #2:

Remarks to the Author:

The authors report the development of a self-healing adhesive conductive hydrogel shown to electro-couple with infarcted tissues. The concept is very relevant for the field of bioelectronics; however from the manuscript it is not clear how this particular CAHP system compares to other self-healing and self-adhesive conductive hydrogels reported in the literature, and reviewed for example in *Adv. Mater.* 2022, 34, 2108932. The major concern is the lack of testing a PVA hydrogel as the control in the MI model. After all, the CAHP hydrogels exhibit modest improvements in its electronic properties and ionic conduction, which will be present in a PVA hydrogel, has been previously shown to exhibit an effect on MI tissue, albeit mechanical. Another concern is the claim that the f-PANI is self-doped. The authors do not provide a scientific proof to support this claim, and their UV spectra indicate the opposite, a reduced f-PANI with b-PANI and PANI having their absorption increasing at $\lambda > 780\text{nm}$, indicative of a possible development of a bipolaron in these two polymers (not the case for f-PANI). Further, the lack of self-doping is supported by the very low conductivity reported for all 3 polymers. If the authors maintain the self-doping mechanism, then they should elaborate and discuss what is causing the self-doping. Other minor concerns include:

Page 13, line 236: what do the authors mean by "the post-gel with complete crosslinking (>10 min) was unable to adhere to other nontarget tissues (Fig. 3K)." this is not clear from Fig 3K.

How was CAHP before gelation imaged in SEM? Before gelation, one would expect it to be a solution. If it is a solution, how can it show this porous organised morphology? This part needs clarification. Apparently passing the pre-gel solution in the applicator (Fig. S4) forms a hydrogel. If it is the case, then gelation is occurring, and the authors should not refer to it as a pre-gel.

What is the healing or recovery time of the hydrogel and how does it compare to other self-healing conductive hydrogels reported in the literature?

In its current form, the manuscript does not outline clearly why this CAHP is an outstanding candidate compared to published systems. The in vivo experiments would benefit significantly from repeating all tests with a PVA hydrogel as a control. If f-PANI is self-doped, a clear mechanism and supporting data should be presented, how is it self-doped and a justification of the very low electronic properties exhibited by this hydrogel should be discussed.

Point-by-point response to the reviewers' comments

Reviewer #1:

The authors report a novel two-part tissue-adhesive conductive hydrogel-based cardiac patch for cardiac monitoring and electrocoupling treatment of myocardial infarction. The benchtop characterization and analysis of the material design, chemistry, adhesion performance, and electrical and electrochemical properties were conducted. The pre-clinical evaluation of the cardiac monitoring and electrocoupling treatment efficacy of MI was performed based on a small animal model. The proposed material and its application for in vivo cardiac monitoring and MI treatment were well characterized based on the comprehensive set of experiments and corresponding analyses for chemical, mechanical, electrical, and biological aspects of the material as well as the target application. The data and presentation are of high quality and comprehensive for the early R&D stage research. The reviewer finds the current work would be a meaningful addition to the community although there are several minor suggestions that the authors can address before the publication. The detailed comments are provided below:

Response: Thank you for your favorable comments.

1. The two-part injectable in situ cross-linkable tissue adhesives are widely studied and even commercialized in various compositions and target applications including epicardial interfacing. Although specific chemistry of materials might differ with the current work, the overall sol-gel transition (non-adhesive after gelation) and adhesion

mechanism (initial physical contact of epicardial surface -> formation of covalent/physical bonds + mechanical interlocking upon crosslinking/gelation) of those existing solutions are essentially comparable to the current work. In this sense, the reviewer finds a bit confused about what the authors mean by ‘chronological’ and ‘spatiotemporal’ greatly emphasized throughout the work given it is really not really unique feature of the proposed solution. It would be great if the authors can clarify how the proposed solution may uniquely differ from existing two-part tissue adhesive hydrogels or make the discussion more technically focused than conceptual selling.

Response:

Thank you for your comments. Our outstanding innovation is the construction of temporally controllable embedded adhesion hydrogels using molecular modifications. According to your suggestions, we clarified what makes the proposed solution unique from existing two-part tissue adhesive hydrogels, including embedded interlocking structures and a clearer chronology adhesion.

(1) Embedded interlocking adhesion with myocardium

There are two salient features of the adhesion mechanism of current two-part tissue adhesives: 1) the precursor liquid can more easily drain the water layer that is resistant to adhesion on the tissue surface, and 2) the polymer forms interfacial linkage interactions with the groups (carboxyl, amino, sulfhydryl groups, etc.) at the tissue surface. As shown in Fig. R1, the embedded interlocking structure of the prepared CAPH in this study is distinguished from the interfacial linkage interaction of the conventional two-part hydrogel. Indeed, water drainage alone is insufficient for

hydrogel adhesion on the heart surface. The anti-adhesive pericardial fluid on the cardiac surface is composed of water and oil, and the constant exudation of oil from the tissue erodes the adhesion interface. Designing wet-adhesion hydrogels for visceral tissues requires rapid breakthrough of the anti-adhesion layer of the oil-water fluid film. In addition, the number of bondable groups at the tissue interface is limited, which results in a relatively low adhesion threshold and poor adjustability of the adhesion strength. Therefore, we prepared functionalized polyaniline (f-PANi) with both hydrophilic and lipophilic properties by side chain modification of conducting polymers. This amphiphilic design allows the conducting polymer to rapidly break through the oil- and water-containing liquid membrane on the epicardium and penetrate into the pericardial interface. In contrast, conventional polymers lack the ability to rapidly and proactively infiltrate at the molecular level. Importantly, the side chains of carboxylic and boronic acids on f-PANi can form noncovalent hydrogen bonds and covalent borate ester bonds with PVA, improving both the interfacial and internal strength. Unlike other two-part adhesive hydrogels, f-PANi is inserted inside the tissue and induces the embedded interlocking structure. Thus, as shown in Fig. R1, conventional two-part adhesive hydrogels are anchored by interfacial bonding with tissues, whereas CAHPs are mutually embedded with tissues. The invasive topology is a key factor in the adhesion toughness and threshold enhancement of CAHP.

Fig. R1 Schematic diagram of traditional adhesion hydrogel and CAHP. While conventional adhesive hydrogels are attached to groups on the tissue, CAHP can form embedded interlocking adhesions with the tissue.

(2) Clear chronology adhesion

Although the adhesion operations of CAHP are similar to those of the two-part tissue adhesive hydrogel, the concept and effect of chronological adhesion is clearer. For example, Liu *et al.* utilized the ligand interactions between Fe^{3+} and catechol-modified gelatin to trigger the in situ formation of epicardial patches, and the binding of catechol to nucleophilic groups (amino and sulfhydryl groups) at the tissue interface could enhance interfacial adhesion.¹ Christman *et al.* developed a PEG-based hydrogel cross-linked by dynamic covalent oxime bonds, in which catechol molecules and aldehyde groups reacted with amines on the epicardial membrane to enhance interfacial adhesion with the epicardium.² These two-part adhesive hydrogels coated the tissue surface in the initial state and formed self-adhesive patches after complete gelation. However, these hydrogels may not be immune to the risk that the strong adhesion moiety, catechol, also binds to other tissues, resulting in nonspecific adhesion. To

address this issue, Rogers *et al.* applied chitosan/EDC/NHS anchors directionally to the tissue surface and selectively triggered the Alg/Ca²⁺/PEG-DA hydrogel to bind firmly to the target tissue via an amidation reaction. No anchoring agent (chitosan/EDC/NHS) was applied to the contralateral hydrogel, which avoided nonspecific adhesion of the hydrogel to other tissues.³ However, the biocompatibility of small molecule anchors and UV curing needs to be further improved. Therefore, further improvement of the adhesion strategy is needed to achieve in situ gelation of epicardial patches with asymmetric adhesion under mild conditions. f-PANi proactively penetrates into the tissue interface in the initial gel state and is locked by a cross-linking network in the completed gel state. Hydrogen bonds and borate ester bonds occurred simultaneously at the tissue interface and within the hydrogel, increasing cohesion and adhesion within 10 min. After complete gelation, CAHP failed to adhere to other tissues because it could no longer form an embedded interlocking structure. Thus, CAHP can simultaneously achieve firm anchoring to target tissues and reject nonspecific adhesion to other tissues by controlling the complete gelation time alone. We have utilized molecular modifications of conducting polymers to achieve time-mediated changes in adhesion, thus emphasizing the concept of “chronological” or “spatiotemporal” adhesion.

The related contents have been added to the revised manuscript to highlight the adhesive advantages of the prepared CAPH.

Line 7–17, Page 4: The physical interactions^{32–34} and chemical linkages^{35–37} between the hydrogel and tissue can improve adhesion strength and interfacial toughness, but the limited functional groups in the epicardium surface result in low adhesive

thresholds³⁸. Additionally, the oil-water barrier on the epicardium surface suppresses the penetration and invasion of polymeric chain segments³⁹, which causes the construction of embedded topological connections and mechanical interlocking networks between hydrogels and myocardium to be a significant challenge. Critically, to prevent nonspecific adhesion of hydrogel patches to other tissues^{40–42}, there is an urgent requirement to modulate the gelation process, network structure, and viscoelasticity on the time scale to achieve spatiotemporally targeted and robust adhesion.

Line 15–17, Page 5: Compared with other two-part adhesive hydrogels^{13,30}, CAHP selectively forms embedded interlocking adhesion structures with tissues by simple manipulation of gelation time.

2. Based on Fig. 3E, the failure mode is cohesive failure. In this case, the adhesion performance may largely dependent upon the fracture toughness of the CAHP but it was not separately measured in the work. It would be informative to provide a fracture toughness of the CAHP as it may directly correlate with its adhesion performance.

Response:

Thank you for your comments. The fracture strength and fracture toughness of CAHP (Supplementary Fig. 5) were investigated according to your suggestion. In detail, we reanalyzed the stress–strain curves in the uniaxial tensile tests (Fig. 4e), counted the fracture strength (Supplementary Fig. 5a), and calculated the fracture toughness (Supplementary Fig. 5b). As a result, when the solid content of CAHP was adjusted from 8% to 12%, the fracture strength increased from 24.83 ± 1.38 kPa to 62.03 ± 6.94

kPa, and the fracture toughness improved from $52.57 \pm 5.18 \text{ kJ}\cdot\text{m}^{-3}$ to $79.57 \pm 8.30 \text{ kJ}\cdot\text{m}^{-3}$ (Supplementary Fig. 5). The increase in the content of PVA and f-PANi induced an increase in the cross-linking density of CAHP, which improved the cohesion of CAHP to resist tensile fracture and increased the ability to store elastic potential energy. The increase in fracture toughness improved the adhesive strength of CAHP for interfacial interlocking and internal cohesion. Thus, with the increase in PVA and f-PANi, the adhesion strength of CAHP increased from 4.84 kPa to 13.65 kPa, and the adhesion toughness improved from $166.2 \text{ J}\cdot\text{m}^{-2}$ to $443.4 \text{ J}\cdot\text{m}^{-2}$. As expected from the comments, the fracture toughness and adhesion performance of CAHP were positively correlated. The analysis of CAHP fracture strength and toughness refines our interpretation of the adhesion mechanism. New Figures were added to the Supplementary Information, and the relevant results and methods were added to the revised manuscript.

The revisions are also pasted below:

Line 20, Page 13–Line 3, Page 14: The increase in cross-linking density improved the tensile fracture strength and toughness of CAHP (Supplementary Fig. 5a, b), which enhanced the ability to maintain interfacial interlocking and resist cohesive failure. As the solid content of CAHP was increased from 8% to 12%, the adhesion strength and toughness of CAHP increased by 2.67 and 2.82 times, respectively.

Line 22, Page 35–Line 2, Page 36: The fracture strength was the stress when the CAHP was destroyed by stretching. Fracture toughness was calculated from the integral area of the tensile stress–strain curve.

Supplementary Fig. 5 | The fracture strength (a) and fracture toughness (b) of CAHP in uniaxial tensile testing ($n = 3$).

3. It is unclear that whether all mechanical characterizations were conducted with as-prepared/applied CAHP or fully equilibrated (or in equilibrium swollen state in physiological environment) states. As the CAHP only contains dynamic covalent bonds (with relatively lower bond energy) and physical bonds, the stability and configurational changes (swelling, mechanical properties, etc) would be critical to actual in vivo application and efficacy. The authors may need to provide clarification on the test conditions as all of these tests would best be done in fully swollen/equilibrated conditions not as-prepared/applied states.

Response:

We agree with your comments that the mechanical properties of the prepared CAHP in the swelling state are more in line with the application in vivo. Therefore, all mechanical property tests were performed after a fully equilibrated state of CAHP in PBS solution in the original manuscript. To clearly explain the mechanical

characterizations, more detailed test conditions were added to the revised manuscript. In addition, to explain the mechanical stability of dynamic CAHP in a physiological environment, we investigated the swelling property and the mechanical property before swelling. The detailed method and results are as follows.

CAHPs were immersed in PBS solution (10 mM) and incubated at 37°C for 96 h. The swelling process of CAHPs was recorded using the weighing method. The medium used for CAHP preparation was PBS, so CAHPs were isotonic and did not swell significantly in PBS. CAHP-8%, CAHP-10% and CAHP-12% reached swelling equilibrium within 24 h. The swelling equilibrium rate of CAHP-12% ($11.4 \pm 2.5\%$) was significantly lower than that of CAHP-8% ($24.7 \pm 3.1\%$) and CAHP-10% ($21.7 \pm 3.4\%$). An increase in solid content from 8% to 12% increased the cross-linking density and decreased the swelling rate. After reaching swelling equilibrium, the mechanical properties of the prepared CAHPs were investigated. The related contents were added to the revised manuscript and *Supplementary Materials*, and they are also pasted below:

Line 8–11, Page 15:

CAHPs isotonic with PBS showed insignificant swelling behavior and reached equilibrium within 12 h (Supplementary Fig. 9a). As the solid content was increased from 8% to 12%, the cross-linked density increased, and the equilibrium swelling rate decreased from $24.7 \pm 3.1\%$ to $11.4 \pm 2.5\%$ (Supplementary Fig. 9b).

Line 14–15, Page 35:

Mechanical testing of CAHP was performed after swelling equilibrium in 10 mM

PBS.

Supplementary Materials:

Swelling tests. The swelling behaviors of the CAHP were investigated by weight analysis. The prepared CAHPs were soaked in PBS solution for 96 h and weighed at the set time. The swelling rate was calculated using supplementary equation (1):

$$\text{Swelling rate} = \frac{W_t - W_0}{W_0} \times 100 \quad (1)$$

where W_t and W_0 are defined as the swollen weight and original weight of the CAHP hydrogel samples, respectively.

Supplementary Fig. 9 | (a) The swelling performance of CAHP in PBS solution for 96 h. **(b)** Equilibrium swelling rate of CAHPs ($n = 3$).

In addition, we performed a uniaxial tensile test of CAHP before swelling to compare the effect of swelling properties on mechanical properties. As shown in Fig. R2, CAHP showed no significant change in the low-strain region ($< 100\%$) before and after swelling in the PBS solution. However, the CAHPs were more prone to yielding at high strains after swelling. The results showed that the fracture elongation of CAHP increased by about 19% and the fracture strength decreased by about 22% after swelling

equilibrium. However, the strain operating range of CAHP as a micro strain sensor is low (25%). Therefore, swelling has a negligible effect on the mechanical properties at low strains. In addition, the mechanical properties after swelling equilibrium are more precise for their application in vivo according to your suggestions. Therefore, the related mechanical data before swelling were not added in the revised manuscript.

Fig. R2. Tensile properties of CAHP before and after swelling. Tensile stress–strain curves (a), fracture elongation (b), and fracture strength (c) of CAHP before and after swelling.

4. The authors correlate data in Fig. 4H to the anti-fatigue property of the CAHP. However, small hysteresis in cyclic tensile tests represents resilience, not anti-fatigue property as the authors seems didn't provide specific fatigue threshold data. Related, the data shows that the material does not have mechanical dissipation but it is confusing how the material can be tough & provide good adhesion performance. What's the exact mechanism of material robustness and good adhesion performance?

Response:

Thank you for your comments. It is indeed ambiguous to associate Fig. 4H with fatigue resistance properties. According to your suggestions, we attribute the low

hysteresis and small energy dissipation of CAHP to resilience. In fact, cyclic tensile testing of the CAHP was performed over a range of simulated heart deformations (25%) and frequencies (1.25 Hz). It was demonstrated that the CAHP maintained resilience behavior over a low strain range. However, the mechanical hysteresis of the CAHP would increase if the strain and frequency are increased. In this work, to assess the mechanical adaptability of CAHP for myocardial patch applications, we demonstrated that CAHP maintains resilience behavior in this operating range.

In addition, the mechanical dissipation of Fig. 4h does not fully reflect the energy dissipation in the adhesion behavior. The low mechanical dissipation in Fig. 4h indicates that the CAHP is resilient rather than relaxed under this test condition. Adhesion rupture was subjected to higher strains and stresses than the cyclic tensile test condition. Therefore, how much energy is stored in the CAHP in the adhesion behavior should be mentioned in the energy dissipation of the CAHP in uniaxial stretching (Fig. 3e). Indeed, as you suggested in *Comment 2*, we explain the relationship between the strength and adhesion properties of the prepared CAHP. The increase in the content of PVA and f-PANi induced an increase in the cross-linking density of CAHP, which improved the cohesion of CAHP to resist tensile fracture. The increase in fracture toughness improved the adhesive strength of CAHP for interfacial interlocking and internal cohesion. The fracture toughness and adhesion performance of CAHP were positively correlated.

5. Regarding in vivo treatment of MI, the reviewer is curious whether the authors could

decouple the mechanical support effect by adhering a swollen hydrogel to the epicardium. It has been known that mechanical augmentation by adhesive viscoelastic hydrogel to MI heart can provide a significant repair effect without electrical conductivity (for example, Lin et al, Nature Biomedical Engineering 3, 632-643 (2019)). A rodent MI model is susceptible to mechanical and electrical effect coupled due to small organ size potential volume-occupying effect of the sizable implant to the heart. The reviewer thinks that having a negative control group (similar hydrogel without conducting polymer) maybe helpful to highlight the electrocoupling treatment efficacy alone.

Response:

Thank you for your comments. We agree with your comments that a rodent MI model is susceptible to mechanical and electrical effects. Our previous study (ACS Nano 2022, 16, 16234–16248) showed that the mechanical and electrical properties of cardiac patches were the main factors for cardiac repair. To highlight the electrocoupling treatment efficacy, we prepared a PVA hydrogel without conducting polymers and investigated its effects on cardiac repair as a negative control group. PVA hydrogels were formed by crystallization of chain segments after freeze–thaw operations. PVA hydrogels were implanted in the MI area using a similar implantation method as that used for CAPH for 28 days. During this period, we evaluated the cardiac repair effect of the PVA hydrogel by electrocardiography (Fig. 7), echocardiography (Fig. 7), Masson staining (Fig. 7), and immunofluorescence staining (Fig. 7) and compared its differences with the MI group and CAHP group.

The electrocardiographic results demonstrated that the PVA hydrogels had difficulty modulating MI-induced electrophysiologic dysfunction due to the lack of conductive activity (Fig. 6j, k). Echocardiographic and histologic results suggested that the PVA hydrogel can attenuate the deterioration of cardiac stroke function by limiting ventricular dilatation and inhibiting compensatory hypertrophy (Fig. 7a, d, e), which was consistent with Lin's report (Nat Biomed Eng 2019, 3, 632-643). However, PVA-treated hearts still showed a trend toward decreased cardiac function within 28 days. In contrast, CAHP improved impaired cardiac function in infarcted myocardium to levels close to the normal range within 28 days, which illustrated that CAHP restored electrophysiologically correlated diastolic-systolic function through electrical coupling. Owing to the potential volume-occupying effect, the PVA hydrogel increased ventricular wall thickness in the infarcted region and inhibited compensatory hypertrophy of cardiomyocytes in the infarcted border region (Fig. 7a, b, c), which originated from the mechanical compensatory effect of PVA hydrogel patches on the left ventricle. However, there were no significant differences between the PVA group and the MI group in terms of infarct size, vessel formation, myocardial retention, and electrically coupled-related Cx43 expression. Therefore, electrocoupling is one of the main factors in improving cardiac repair effects. The new content has been added and labeled in the revised manuscript and Supplementary Materials. In addition, we have changed the order of some images and text for better understanding.

The revisions are also pasted below:

Line 8–13, Page 26:

On postoperative day 3, ECGs of all groups showed markedly elevated ST segments, suggesting acute myocardial injury (Fig. 6j). Subsequently, pathological Q-wave deepening, T-wave inversion, and prolongation of the QRS duration were observed in the MI and PVA groups, suggesting that electrically decoupled fibrosis affected depolarization and repolarization processes.

Fig. 6. Cardiac mechanophysiology monitoring and electrocoupling treatment by CAHP. **j** Representative ECGs for rats in the MI, PVA, and CAHP groups at 3, 14, and 28 days postoperatively. **k** QRS interval duration ($n = 3$). (n.s.: no significant difference, * $p < 0.05$, *** $p < 0.01$, **** $p < 0.001$)

Line 14–18, Page 27:

In contrast, the PVA hydrogel was unable to assist myocardial systole-diastole because of the lack of conductive activity and was passive in limiting geometric expansion of the infarcted myocardium. In addition, cardiac function tended to decline over 28 days in the MI and PVA groups, whereas CAHP improved cardiac pulsatile function.

Line 8, Page 29–Line 7, Page 30:

To explore the repair mechanisms, the infarct areas in the MI, PVA, and CAHP groups were analyzed by immunofluorescence staining. The CAHP group exhibited more extensive areas of positive cardiac troponin T (cTnT) and α -actin expression and more α -SMA-labeled vascular lumens (Fig. 7f and Supplementary Fig. 24a). In contrast, the PVA hydrogel had no significant modulating effect on vascular regeneration and electrical coupling. Notably, numerous vessels were observed in the epicardium surrounding the CAHP, restoring the blood supply in the ischemic region to prevent further disease deterioration. Compared with the PVA hydrogel, the CAHP increased the blood vessel density of the MI area from $6.1 \pm 1.3\%$ to $10.2 \pm 1.5\%$ ($p < 0.001$; Fig. 7g). The expression level of connexin 43 (Cx43) in the infarcted area was low in the MI and PVA groups, suggesting that intercellular electrical communication was greatly weakened. CAHP markedly improved Cx43 expression levels and promoted the myocardial electrical signaling pathway and electric contraction coupling (Fig. 7f and Supplementary Fig. 24b). In addition, cardiomyocytes have an inferior regenerative capacity and only compensate for quantity loss by enlarging their volume size. PVA hydrogel alleviated pathological myocardial hypertrophy through mechanical support ($p < 0.05$), whereas CAHP further reduced myocardial size to $349.5 \pm 68.0 \mu\text{m}^2$ from $502.1 \pm 70.5 \mu\text{m}^2$ of PVA hydrogel ($p < 0.01$; Fig. 7f, h). Thus, compared with nonconductive PVA hydrogel, electrically coupled therapeutic CAHP could more effectively inhibit ventricular remodeling, reduce fibrotic scarring, promote vascular regeneration, and synergistically repair myocardial morphology and function.

Fig. 7. Myocardial repair effects of CAHP. **a** Masson staining for collagen (blue) and muscle (red) and echocardiography imaging in the MI, PVA and CAHP groups. The solid white line annotates the left ventricular wall thickness, and the yellow dashed line marks the outline of the CAHP. **a, b** Quantitative analysis of the wall thickness (**b**) and infarcted area (**c**) in the MI, PVA, and CAHP groups ($n = 5$). **d, e** The EF (**d**) and SV (**e**) of the left ventricle after 3, 7, 14, and 28 days postoperatively in the MI and CAHP groups ($n = 3$). **f** Representative immunofluorescence staining of the infarct region in the MI, PVA, and CAHP groups for cardiac markers (cTnT and α -actin), vascularization marker (α -SMA), intercellular electrical contraction coupling marker (Cx43),

cardiomyocyte profiles (WGA), and nuclei (DAPI). **g, h** Quantitative analysis of blood vessel density (**g**) and myocyte size (**h**) ($n = 5$). (n.s.: no significant difference, $*p < 0.05$, $**p < 0.01$, $***p < 0.001$)

Reviewer #2:

Major concerns:

1. The authors report the development of a self-healing adhesive conductive hydrogel shown to electro-couple with infarcted tissues. The concept is very relevant for the field of bioelectronics; however, from the manuscript it is not clear how this particular CAHP system compares to other self-healing and self-adhesive conductive hydrogels reported in the literature, and reviewed for example in Adv. Mater. 2022, 34, 2108932.

Response:

Thank you for your comments. We explain the differences between the CAHP and the reported self-healing and self-adhesive hydrogels in the revised manuscript, including Li's report (Adv Mater 2022, 34, 2108932).⁴ An electrical self-healing test was performed to further explain the self-healing of the CAHP.

(1) Self-healing properties

Compared with the reported self-healing conductive hydrogels, the self-healing properties of CAHP are characterized by two outstanding features: 1) it does not depend on any external stimuli, and 2) the conductive component is involved in network reconstruction. The self-healing of many hydrogels is conditional and must be triggered in response to stimuli such as humidity, temperature, and light. The self-healing of

CAHP is mediated by noncovalent hydrogen bonding and dynamic covalent borate ester bonds without any additional stimuli, which allows CAHP to automatically adapt to the mechanical environment of dynamic tissues. In addition, previously self-healing groups were modified in other nonconductive polymers within the hydrogel, which resulted in the conducting polymer being out-of-phase in the self-healing network. In this work, self-healing groups were covalently modified on the conjugated backbone of the conducting polymer, allowing the conductive component to directly participate in the network remodeling process, which is a key factor in accelerating the self-healing rate. According to step-strain rheological scanning tests, we calculated the mechanical self-healing efficiency of the CAHP after multiple shear disruptions to be approximately 98% (Fig. 4j). In addition, the CAHP was connected to a circuit to monitor the current changes when the CAHP was cut and re-contacted. The results showed that the severed CAHP could rapidly regenerate the conductive channels to restore the current to the original level after re-contact and the electrical healing efficiency of CAHP was about 100% (Supplementary Fig. 11). As a result, CAHP can rapidly and automatically restore the damaged mechanical and electrical properties to almost the original level compared to the reported self-healing conductive hydrogels.

(2) Self-adhesive properties

Traditionally, the adhesive formation of hydrogels on tissue depends on the interactions between adhesion groups (e.g., aldehyde groups, catechols, NHS, etc.) in hydrogels and functional groups (e.g., amino, carboxyl, sulfhydryl groups, etc.) on the tissue. However, the number of functional groups on the tissues is limited, and such

interfacial connected structures suffer from the defects of a low adhesion threshold and poor controllability. In this work, the molecular modification strategy allows rapid penetration of the conducting polymer into the tissue and in situ covalent/noncovalent cross-linking of the CAHP. Compared with previously reported conductive hydrogels, CAHP realizes an embedded interlocking structure with tissues and has the following advantages. 1) The invasive adhesive structure improved the adhesion toughness, which is higher than that of the hydrogels that have been reported to adhere to the epicardium.^{1,4,5} 2) Conventional conducting polymers within hydrogels have only a weak physical interaction with tissue, whereas f-PANi is directly involved in the adhesion process through covalent/noncovalent interactions, which facilitates electrocoupling with myocardial tissues across the epicardium. 3) Conventional conductive hydrogels have pursued the enhancement of adhesion performance while neglecting the possibility that strong adhesion groups may produce nonspecific adhesion with other tissues in the thoracic cavity. CAHP in contact with the epicardium in the initial gel state can produce strong adhesion after in situ cross-linking and is unable to form interfacial adhesion with other tissues after complete gelation due to restricted f-PANi movement by the cross-linked network. Thus, tissue adhesion of CAHP is chronological and selectively adheres to target tissues at a predicted time and space. In addition, we have added a comparison with other epicardial adhesion hydrogels to the manuscript. Enhanced adhesion thresholds and the chronological adhesion mechanism are emphasized in the manuscript.

These features of CAPH in self-healing properties and self-adhesive properties

compared with previous reports were emphasized in the revised manuscript according to your suggestion. The related contents were also pasted here.

Line13–16, Page 16:

Predominantly, dynamic covalent borate ester bonds can recombine without relying on external stimuli, mediating strong self-healing forces to quickly compensate for material defects. The severed CAHP self-healed quickly after re-contacting and maintained a stable connection after stretching (Supplementary Video 2).

Line 21, Page 16–Line 6, Page 7:

In the step strain between 10% and 400%, CAHPs could be self-healed to their original modulus almost indestructibly (mechanical self-healing efficiency $\approx 98\%$) after stress damage, which was repeatable upon additional strain cycles (Fig. 4j). In addition, the electrical self-healing efficiency in the disconnection-reconnection operation was approximately 100% (Supplementary Fig. 11). Compared to other self-healing conducting polymer hydrogels⁴⁶, f-PANi was able to directly participate in network remodeling, achieving higher mechanical and electrical self-healing efficiency.

Supplementary Materials:

Electrical self-healing test. The CAHP was connected to a circuit with a 3 V potential, and the CAHP was disconnected and reconnected by the blade. Current changes were monitored to further demonstrate the self-recovery of electrical connections within the hydrogel networks. Electrical healing efficiency calculated by the ratio of the current value of the reconnection to the current value before disconnection.

Supplementary Fig. 11 | The current changes of the electrical self-healing process in the disconnect-reconnect cycle of CAHP.

2. The major concern is the lack of testing a PVA hydrogel as the control in the MI model. After all, the CAHP hydrogels exhibit modest improvements in its electronic properties and ionic conduction, which will be present in a PVA hydrogel, has been previously shown to exhibit an effect on MI tissue, albeit mechanical.

Response:

Thank you for your comments. We agree with your comments that a rodent MI model is susceptible to mechanical and electrical effects. Our previous study (ACS Nano 2022, 16, 16234–16248) showed that the mechanical and electrical properties of cardiac patches were the main factors for cardiac repair. To highlight the electrocoupling treatment efficacy, we investigated the effects of PVA hydrogel patches without conducting polymers as a negative control group for cardiac repair. PVA hydrogel patches were formed by crystallization of chain segments after freeze–thaw operations. PVA hydrogels were implanted in the MI area using a similar implantation method as that used for CAPH for 28 days. During this period, we evaluated the cardiac

repair effect of the PVA hydrogel by electrocardiography (Fig. 7), echocardiography (Fig. 7), Masson staining (Fig. 7), and immunofluorescence staining (Fig. 7) and compared its differences with the MI group and CAHP group.

The electrocardiographic results demonstrated that the PVA hydrogels had difficulty modulating MI-induced electrophysiologic dysfunction due to the lack of conductive activity (Fig. 6j, k). Echocardiographic and histologic results suggested that the PVA hydrogel can attenuate the deterioration of cardiac stroke function by limiting ventricular dilatation and inhibiting compensatory hypertrophy (Fig. 7a, d, e). However, PVA-treated hearts still showed a trend toward decreased cardiac function within 28 days. In contrast, CAHP improved impaired cardiac function in infarcted myocardium to levels close to the normal range within 28 days, which illustrated that CAHP restored electrophysiologically correlated diastolic-systolic function through electrical coupling. Owing to the potential volume-occupying effect, the PVA hydrogel increased ventricular wall thickness in the infarcted region and inhibited compensatory hypertrophy of cardiomyocytes in the infarcted border region (Fig. 7a, b, c), which originated from the mechanical compensatory effect of PVA hydrogel patches on the left ventricle. However, there were no significant differences between the PVA group and the MI group in terms of infarct size, vessel formation, myocardial retention, and electrically coupled-related Cx43 expression. Therefore, electrocoupling is one of the main factors in improving the cardiac repair effects. The new content has been added and labeled in the revised manuscript and Supplementary Materials. In addition, we have changed the order of some images and text for better understanding.

The revisions are also pasted below:

Line 8–13, Page 26:

On postoperative day 3, ECGs of all groups showed markedly elevated ST segments, suggesting acute myocardial injury (Fig. 6j). Subsequently, pathological Q-wave deepening, T-wave inversion, and prolongation of the QRS duration were observed in the MI and PVA groups, suggesting that electrically decoupled fibrosis affected depolarization and repolarization processes.

Fig. 6. Cardiac mechanophysiology monitoring and electrocoupling treatment by

CAHP. j Representative ECGs for rats in the MI, PVA, and CAHP groups at 3, 14, and

28 days postoperatively. **k** QRS interval duration ($n = 3$). (n.s.: no significant difference,

* $p < 0.05$, *** $p < 0.01$, **** $p < 0.001$)

Line 14–18, Page 27:

In contrast, the PVA hydrogel was unable to assist myocardial systole-diastole because of the lack of conductive activity and was passive in limiting geometric expansion of the infarcted myocardium. In addition, cardiac function tended to decline over 28 days in the MI and PVA groups, whereas CAHP improved cardiac pulsatile function.

Line 8, Page 29–Line 7, Page 30:

To explore the repair mechanisms, the infarct areas in the MI, PVA, and CAHP groups were analyzed by immunofluorescence staining. The CAHP group exhibited more extensive areas of positive cardiac troponin T (cTnT) and α -actin expression and more α -SMA-labeled vascular lumens (Fig. 7f and Supplementary Fig. 24a). In contrast, the PVA hydrogel had no significant modulating effect on vascular regeneration and electrical coupling. Notably, numerous vessels were observed in the epicardium surrounding the CAHP, restoring the blood supply in the ischemic region to prevent further disease deterioration. Compared with the PVA hydrogel, the CAHP increased the blood vessel density of the MI area from $6.1 \pm 1.3\%$ to $10.2 \pm 1.5\%$ ($p < 0.001$; Fig. 7g). The expression level of connexin 43 (Cx43) in the infarcted area was low in the MI and PVA groups, suggesting that intercellular electrical communication was greatly weakened. CAHP markedly improved Cx43 expression levels and promoted the myocardial electrical signaling pathway and electric contraction coupling (Fig. 7f and Supplementary Fig. 24b). In addition, cardiomyocytes have an inferior regenerative capacity and only compensate for quantity loss by enlarging their volume size. PVA hydrogel alleviated pathological myocardial hypertrophy through mechanical support ($p < 0.05$), whereas CAHP further reduced myocardial size to $349.5 \pm 68.0 \mu\text{m}^2$ from $502.1 \pm 70.5 \mu\text{m}^2$ of PVA hydrogel ($p < 0.01$; Fig. 7f, h). Thus, compared with nonconductive PVA hydrogel, electrically coupled therapeutic CAHP could more effectively inhibit ventricular remodeling, reduce fibrotic scarring, promote vascular regeneration, and synergistically repair myocardial morphology and function.

Fig. 7. Myocardial repair effects of CAHP. **a** Masson staining for collagen (blue) and muscle (red) and echocardiography imaging in the MI, PVA and CAHP groups. The solid white line annotates the left ventricular wall thickness, and the yellow dashed line marks the outline of the CAHP. **a, b** Quantitative analysis of the wall thickness (**b**) and infarcted area (**c**) in the MI, PVA, and CAHP groups ($n = 5$). **d, e** The EF (**d**) and SV (**e**) of the left ventricle after 3, 7, 14, and 28 days postoperatively in the MI and CAHP groups ($n = 3$). **f** Representative immunofluorescence staining of the infarct region in the MI, PVA, and CAHP groups for cardiac markers (cTnT and α -actin), vascularization marker (α -SMA), intercellular electrical contraction coupling marker (Cx43),

cardiomyocyte profiles (WGA), and nuclei (DAPI). **g, h** Quantitative analysis of blood vessel density (**g**) and myocyte size (**h**) ($n = 5$). (n.s.: no significant difference, $*p < 0.05$, $**p < 0.01$, $***p < 0.001$)

3. Another concern is the claim that the f-PANI is self-doped. The authors do not provide a scientific proof to support this claim, and their UV spectra indicate the opposite, a reduced f-PANI with b-PANI and PANI having their absorption increasing at $\lambda > 780$ nm, indicative of a possible development of a bipolaron in these two polymers (not the case for f-PANI). Further, the lack of self-doping is supported by the very low conductivity reported for all 3 polymers. If the authors maintain the self-doping mechanism, then they should elaborate and discuss what is causing the self-doping.

Response:

Thank you for your comments. We further demonstrate the doping structures of the three conducting polymers by UV-vis spectroscopy and new XPS experiments, and carefully analyze the relationship between these data and doping. Combining your comments, we think that the proof of self-doping in the manuscript is immature. The theory of self-doping may need to be systematically investigated in the future. Therefore, we only describe the relationship between the doped structures and the conductivity of the three conducting polymers in the manuscript and do not emphasize that they are self-doped. The detailed analysis is listed below.

In the UV-vis spectra (Fig. 2j), the shoulder peak at 300 nm is attributed to π - π^* . The absorption peak representing the polarization transition of f-PANi (445 nm) is

significantly red-shifted compared to PANi and b-PANi (414 nm), resulting in the easy excitation of electrons in f-PANi and enhanced mobility in the conjugated system. However, the characteristic peak of f-PANi disappears at the bipolaron band transitions (>780 nm). Therefore, the doping structure is complex and cannot be explained by UV-vis spectroscopy alone. In addition, the doping structure of polyaniline also needs to consider the percentage of quinone rings to benzene rings in the conjugated structure. As shown in the FT-IR spectra (Supplementary Fig. 1), the absorption band at 1493 cm^{-1} represents the amine-based benzene ring, whereas the absorption band at 1575 cm^{-1} represents the imine-based quinone ring.⁶ The ratio of the relative intensities of quinoid to benzenoid ring modes (I_Q/I_B) in PANi (0.829) was smaller than that of b-PANi (0.976) and f-PANi (1.053). The I_Q/I_B is close to 1, implying that they are in the redox intermediate state. In addition, the peak position of the benzene ring in f-PANi is blue shifted to 1510 cm^{-1} from 1493 cm^{-1} of PANi, which may be related to the fact that the carboxylic acid and boronic acid side chains provide protons to the main chain to provide more quinone conjugated structures.

To resolve the effect of carboxylic acid and boronic acid side chains on the doped structure, we fit the N1s peaks in the XPS spectra of the three conducting polymers. According to the peak positions of amines (398.5 eV), imines (399.2 eV), and protonated imines (400.8 eV), the proportions of the three structures in the conducting polymers are analyzed.⁷ Importantly, the content of protonated amines in f-PANi (22.39%) and b-PANi (19.63%) is larger than that of PANi (11.85%) because the borate acid and carboxylic acid side chains provide more protons to the imine on the backbone.

The reasons for the low conductivity of conducting polymers are nonacid treatments. For biomedical considerations, we did not use exotic acidic dopants (e.g., HCl, H₂SO₄) to treat the conducting polymer film, which may have contributed to the lower conductivity of f-PANi compared to other acid-doped conducting polymers.⁸

According to these results, we revised the relevant content in the manuscript.

Line 13, Page 9–Line 2, Page 10:

The UV absorption peak representing the polarization transitions is significantly red-shifted to 445 nm of f-PANi from 414 nm in PANi and b-PANi, which leads to easy electron excitation in f-PANi and enhanced mobility in the conjugated system (Fig. 2j). In addition, the relative intensity ratios of the quinone to the benzene ring in b-PANi and f-PANi are closer to 1, suggesting a redox intermediate state⁴³ (Supplementary Fig. 1). The FTIR absorption band of the benzene ring in f-PANi is blue-shifted to 1510 cm⁻¹ from 1493 cm⁻¹ in PANi, implying that f-PANi tends to have a more quinone-conjugated structure. Importantly, the content of protonated imines (–NH⁺=C–) in f-PANi (22.39%) and b-PANi (19.63%) was greater than that in PANi (11.85%), suggesting that the ionized side chains provide more protons for the imines in the backbone⁴⁴ (Supplementary Fig. 2a–d).

Supplementary Fig. 1 | FT-IR spectra of PANi, b-PANi and f-PANi.

Supplementary Fig. 2 | XPS of N1s for PANi (a), b-PANi (b), and f-PANi (c). (d) The doping structures of conducting polymers.

Other minor concerns include:

1. Page 13, line 236: what do the authors mean by “the post-gel with complete

crosslinking (>10 min) was unable to adhere to other nontarget tissues (Fig. 3K).” this is not clear from Fig 3K.

Response:

Thank you for your comments. The description of Fig. 3k was unclear in the original manuscript. In fact, we observed interfacial binding between the tissue and CAHP by H&E staining. As shown in Fig. 3k, the polymer in CAPH could rapidly penetrate into the epicardium before the gelation point and then form a mechanically interlocking structure with the tissue through in situ cross-linking, demonstrating embedded interfacial anchoring. CAHPs lost their strong adhesion behavior upon contact with tissue after complete gelation. We have revised the original expression for the understanding of reviewers and readers.

Line 13–16, Page 13:

Hematoxylin & Eosin staining images showed that initial gel loading enabled CAHP to form an embedded interfacial anchorage with the epicardium, whereas CAHP in the complete gel state no longer adhered to tissues due to the absence of mechanical interlocking structures (Fig. 3k).

2. How was CAHP before gelation imaged in SEM? Before gelation, one would expect it to be a solution. If it is a solution, how can it show this porous organised morphology? This part needs clarification. Apparently passing the pre-gel solution in the applicator (Fig. S4) forms a hydrogel. If it is the case, then gelation is occurring, and the authors should not refer to it as a pre-gel.

Response:

Thank you for your comments. To prepare SEM samples of CAHP in the initial state, the PVA solution and the f-PANi solution were mixed via an applicator and injected into liquid nitrogen within 10 s. The fast-growing ice crystals rapidly immobilized the initially formed network structure. The initial gel of CAHP was freeze-dried, and the internal structure was observed by SEM. We have refined the description of the SEM tests in the Supplementary Material and revised the content in the manuscript.

In addition, the descriptions of the CAHP state before and after the gelation point ($G' = G''$) were ambiguous in the original manuscript. At the initial stage, CAHP rapidly formed a physical cross-linked structure through hydrogen bonding interactions between PVA and f-PANi. The gelation reaction already occurs after solution mixing. However, physically cross-linked hydrogels are weak and loose in structure. Therefore, we changed the term of CAHP state before the gelation point from "pre-gel" to "initial gel" according to your suggestion. The term of CAHP state after the gelation point was changed from "post-gel" to "completed gel".

Supplementary Material:

Morphology characterization. To obtain the network structure of CAHP in the initial gel state, PVA solution was mixed with f-PANi solution and injected into liquid nitrogen within 10 seconds. The rapidly growing ice crystals quickly immobilized the initially formed network structure. In addition, the CAHP in the complete gel state was frozen

after gelation was carried out for 10 min. The samples were freeze-dried and sprayed with gold. The pore structures of CAHP in the initial and complete gelation and the interfacial morphology between CAHP and myocardium were characterized by scanning electronic microscopy (SEM, S-4800, Hitachi, Japan).

3. What is the healing or recovery time of the hydrogel and how does it compare to other self-healing conductive hydrogels reported in the literature?

Response:

Thank you for your comments. To assess the healing time of CAHP, the CAHP was cut and then re-contacted. As shown in Supplementary Video 2, the CAHPs reconnected within 1 s. In contrast to the stimulus-responsive healing hydrogels (dependent on humidity, temperature, light, etc.), the self-healing of CAHP was mediated by dynamic noncovalent hydrogen bonds and covalent borate ester bonds without any additional stimulus. In addition, CAHP has a rapid healing time compared to the reported self-healing conductive hydrogels. The reported self-healing times of self-healing conductive hydrogels are all longer than 15 s.⁹ For example, the PEDOT/PHEA-PSS hydrogel can self-heal after 24 h at a certain pressure by coordination bonding and hydrogen bonding.¹⁰ Renewable Schiff bases and hydrogen bonding allowed PPy/alginate-gelatin hydrogel to heal after 40 min.¹¹ Conductive PANi/CNF/PVA hydrogels based on borate bonds accomplished a faster self-healing process (~15 s).¹² CAHP has a faster healing time (~1 s) and maintains stable connections under immediate stretching (Supplementary Video 2). Previously,

functional groups with self-healing effects were used to modify the nonconductive polymers within the hydrogel, which resulted in the conducting polymer being out-of-phase in the self-healing network. In this work, borate side groups were covalently modified on the conjugated backbone of the conducting polymer, allowing the conductive component to directly participate in the network remodeling process, which is a key factor in accelerating the self-healing rate. We also added movies of healing times and comparisons with the literature in the manuscript.

Line 3–5, Page 16:

The severed CAHP self-healed quickly after recontacting and maintained a stable connection after stretching (Supplementary Video 2).

In its current form, the manuscript does not outline clearly why this CAHP is an outstanding candidate compared to published systems. The *in vivo* experiments would benefit significantly from repeating all tests with a PVA hydrogel as a control. If f-PANI is self-doped, a clear mechanism and supporting data should be presented, how is it self-doped and a justification of the very low electronic properties exhibited by this hydrogel should be discussed.

Response:

Thank you again for your comments. In response to these comments, we have provided additional experiments and detailed discussions. Point-by-point responses have been provided for the above major concerns **Comments 1–3** and other minor concerns. Your suggestions are very helpful for us to improve our manuscript.

Reference

1. Liang S, *et al.* Paintable and rapidly bondable conductive hydrogels as therapeutic cardiac patches. *Adv Mater* **30**, 1704235 (2018).
2. Fujita M, *et al.* Preventing post-surgical cardiac adhesions with a catechol-functionalized oxime hydrogel. *Nat Commun* **12**, 3764 (2021).
3. Yang Q, *et al.* Photocurable bioresorbable adhesives as functional interfaces between flexible bioelectronic devices and soft biological tissues. *Nat Mater* **20**, 1559-1570 (2021).
4. Lin X, *et al.* A viscoelastic adhesive epicardial patch for treating myocardial infarction. *Nat Biomed Eng* **3**, 632-643 (2019).
5. Yu C, *et al.* An intrapericardial injectable hydrogel patch for mechanical-electrical coupling with infarcted myocardium. *ACS Nano* **16**, 16234-16248 (2022).
6. Deore BA, Yu I, Freund MS. A switchable self-doped polyaniline: Interconversion between self-doped and non-self-doped forms. *J Am Chem Soc* **126**, 52-53 (2004).
7. Shi H-Y, Ye Y-J, Liu K, Song Y, Sun X. A long-cycle-life self-doped polyaniline cathode for rechargeable aqueous zinc batteries. *Angew Chem Int Edit* **57**, 16359-16363 (2018).
8. Imae I, Krukiewicz K. Self-doped conducting polymers in biomedical engineering: Synthesis, characterization, current applications and perspectives.

Bioelectrochemistry **146**, 108127 (2022).

9. Li Y, Zhou X, Sarkar B, Gagnon-Lafrenais N, Cicoira F. Recent progress on self-healable conducting polymers. *Adv Mater* **34**, 2108932 (2022).
10. Ding XY, Jia RP, Gan ZZ, Du Y, Wang DY, Xu XW. Tough and conductive polymer hydrogel based on double network for photo-curing 3D printing. *Mater Res Express* **7**, 055304 (2020).
11. Ren K, Cheng Y, Huang C, Chen R, Wang Z, Wei J. Self-healing conductive hydrogels based on alginate, gelatin and polypyrrole serve as a repairable circuit and a mechanical sensor. *J Mater Chem B* **7**, 5704-5712 (2019).
12. Han J, Ding Q, Mei C, Wu Q, Yue Y, Xu X. An intrinsically self-healing and biocompatible electroconductive hydrogel based on nanostructured nanocellulose-polyaniline complexes embedded in a viscoelastic polymer network towards flexible conductors and electrodes. *Electrochim Acta* **318**, 660-672 (2019).

Reviewers' Comments:

Reviewer #1:

Remarks to the Author:

In the revised manuscript, the authors addressed the reviewer's concerns well with the additional data and analysis. The reviewer still has one minor concern that authors may need to correct before publication.

In Supplementary Fig. 5 and related discussion, the experimental quantity measured for 'fracture toughness' is toughness in a unit of $[J/m^3]$. Fracture toughness requires different measurement with different unit $[J/m^2]$.

Reviewer #2:

Remarks to the Author:

The authors have addressed thoroughly the comments of both reviewers, making the manuscript more complete and comprehensive. I recommend its publication in the journal.

Point-by-point response to the reviewers' comments

Reviewer #1:

In the revised manuscript, the authors addressed the reviewer's concerns well with the additional data and analysis. The reviewer still has one minor concern that authors may need to correct before publication.

In Supplementary Fig. 5 and related discussion, the experimental quantity measured for 'fracture toughness' is toughness in a unit of $[J/m^3]$. Fracture toughness requires different measurement with different unit $[J/m^2]$.

Response: Thank you for your comments. We measured the fracture toughness and work by the single-edge notched tensile method according to previous reports.^{1,2} The unit of fracture toughness is $J \cdot m^{-2}$, and the unit of fracture work is $J \cdot m^{-3}$, which adequately reflects the ability of CAHP to resist fracture damage. When the solid content of CAHP was increased from 8% to 12%, the ability to resist crack propagation under tension (Supplementary Fig. 6a–d) was improved, as evidenced by the increase in fracture toughness and fracture work (Supplementary Fig. 6e, f). According to your suggestion, we added methods, results, and discussions of the 'fracture toughness' in the Supplementary Material and revised manuscript.

Supplementary Material:

Fracture tests. Fracture tests were performed on a tensile tester to determine the fracture energy of the CAHPs. The CAHPs were cut into a rectangular shape (40 mm × 10 mm × 2 mm) with a 5 mm initial notch at the midpoint of the long edge. The notched and un-notched specimens of the CAHPs were stretched at a rate of $20 \text{ mm} \cdot \text{min}^{-1}$ to record

the stress (σ) and extension ratio (λ , the ratio of the stretched length to the original length). The fracture strength was the stress when the CAHP was destroyed by stretching. Fracture work (W) was calculated from the integral area of the σ - λ curve of the un-notched samples. Fracture toughness (Γ) was calculated using following the supplementary equation (2):

$$\Gamma = 2 \times \frac{3}{\sqrt{\lambda_c}} \times W(\lambda_c) \times c \quad (2)$$

where λ_c is the extension ratio of the notched sample at the crack fracture. $W(\lambda_c)$ is the integral area of the σ - λ curve for the un-notched samples in the λ_c range. c is the initial length of the crack.

Supplementary Fig. 6 | The fracture energy of the CAHPs. (a) Measurement of the fracture energy by the single-edge notched tensile method. (b–d) Tensile curves of CAHP-8% (b), CAHP-10% (c), and CAHP-12% (d) with and without a notch. (e, f) The fracture toughness (e) and fracture work (f) of CAHPs ($n = 3$).

Line 20, Page 13–Line 4, Page 14:

When the solid content of CAHP was increased from 8% to 12%, the ability to resist crack propagation under tension (Supplementary Fig. 6a – d) was improved, as evidenced by the increase in fracture toughness and fracture work (Supplementary Fig. 6e, f). Therefore, the increase in cross-linking density enhanced the ability to maintain interfacial interlocking and resist cohesive failure, resulting in an increase in the adhesion strength and toughness of CAHP by 2.67 and 2.82 times, respectively.

Reviewer #2 (Remarks to the Author):

The authors have addressed thoroughly the comments of both reviewers, making the manuscript more complete and comprehensive. I recommend its publication in the journal.

Response: Thank you for your favorable comments.

Reference:

1. Zheng SY, *et al.* Molecularly engineered zwitterionic hydrogels with high toughness and self-healing capacity for soft electronics applications. *Chem Mater* **33**, 8418-8429 (2021).
2. Li C, Wang Z, Wang Y, He Q, Long R, Cai S. Effects of network structures on the fracture of hydrogel. *Extreme Mechanics Letters* **49**, 101495 (2021).

Reviewers' Comments:

Reviewer #1:

Remarks to the Author:

The authors have addressed the reviewer's remaining comments in full. The reviewer does not have further concerns and would like to congratulate the authors for their good work.

Point-by-point response to the reviewers' comments

Reviewer #1:

The authors have addressed the reviewer's remaining comments in full. The reviewer does not have further concerns and would like to congratulate the authors for their good work.

Response: We thank the reviewer for the positive response and all valuable suggestions in previous comments to help us improve our manuscript.